# PARTONOMY: Large Multimodal Models with Part-Level Visual Understanding

**Ansel Blume**[1]*, **Jeonghwan Kim**[1]*, **Hyeonjeong Ha**[1], **Elen Chatikyan**[1], **Xiaomeng Jin**[1],
**Khanh Duy Nguyen**[1], **Nanyun Peng**[2], **Kai-Wei Chang**[2], **Derek Hoiem**[1], **Heng Ji**[1]
[1]University of Illinois Urbana-Champaign, [2]University of California Los Angeles
{blume5, jk100, hengji}@illinois.edu

## Abstract

Real-world objects are composed of distinctive, object-specific parts. Identifying these parts is key to performing fine-grained, compositional reasoning—yet, large multimodal models (LMMs) struggle to perform this seemingly straightforward task. In this work, we introduce **PARTONOMY**, an LMM benchmark designed for pixel-level part grounding. We construct **PARTONOMY** from existing part datasets and our own rigorously annotated set of images, encompassing 862 part labels and 534 object labels for evaluation. Unlike existing datasets that simply ask models to identify generic parts, **PARTONOMY** uses specialized concepts (e.g., agricultural airplane), and challenges models to compare objects' parts, consider part-whole relationships, and justify textual predictions with visual segmentations. Our experiments demonstrate significant limitations in state-of-the-art LMMs (e.g., LISA-13B achieves only 5.9% gIoU), highlighting a critical gap in their part grounding abilities. We note that existing segmentation-enabled LMMs (segmenting LMMs) have two key architectural shortcomings: they use special `[SEG]` tokens not seen during pretraining which induce distribution shift, and they discard predicted segmentations instead of using past predictions to guide future ones. To address these deficiencies, we propose **PLUM**, a novel segmenting LMM that uses span tagging instead of segmentation tokens and that conditions on prior predictions in a feedback loop. We find that pretrained **PLUM** outperforms existing segmenting LMMs on reasoning segmentation, VQA, and visual hallucination benchmarks. In addition, **PLUM** finetuned on our proposed *Explanatory Part Segmentation* task is competitive with segmenting LMMs trained on significantly more segmentation data. Our work opens up new avenues towards enabling fine-grained, grounded visual understanding in LMMs. The code and data are publicly available at: https://github.com/AnselBlume/partonomy

## 1 Introduction

Real-world objects can be decomposed into distinctive parts. A banana boat (Fig. 2), for instance, consists of a seating tube, a handle, a hull, and an inflation valve. Such parts characterize each concept, differentiating one object from another. The ability to recognize and distinguish between parts is an important element of holistic object understanding, with applications ranging from explainable object recognition [3, 16, 28, 8, 58], to part-based novel concept design [10], and robotic manipulation [52, 30, 14]. Decomposing objects into their key building blocks allows models to reason about objects at a granular level [16, 29], allowing for more complex and nuanced interactions.

Unfortunately, Large Multimodal Models (LMMs), the backbones of today's multimodal systems, lack strong part recognition abilities [16, 29, 33]. While they perform well on visual reasoning [43, 13, 42] and visual hallucination tasks [23], we find that they are unable to accurately identify

---

* Equal contribution.

39th Conference on Neural Information Processing Systems (NeurIPS 2025).

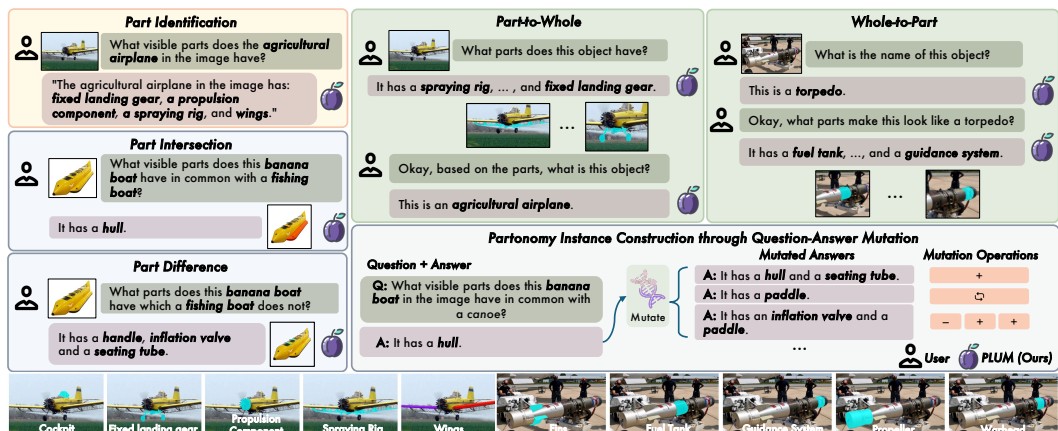

Figure 1: The **PARTONOMY** dataset evaluates LMMs' part understanding through the *Explanatory Part Segmentation* task. Given an input image, a segmentation-enabled LMM selects a textual explanation and generates part segmentation masks which serve as textual and visual rationale for its answer choice. Our question-answer mutation framework generates challenging answer choices by predicting part co-occurrence and by selecting parts from confusable objects.

object parts in an image, occasionally regurgitating object parts memorized from text-only pre-training (e.g. "a fish must have a fin"). Worse, LMMs that can generate segmentation masks, *segmenting LMMs* [19, 37, 41, 49], lack the ability to ground these fine-grained regions despite being trained on part segmentation data. This severely limits LMMs' utility in real-world scenarios that require fine-grained, part-level understanding.

To quantify the part recognition abilities of LMMs, we propose **Explanatory Part Segmentation**, a task that assesses LMMs' ability to recognize object parts, associate objects with their distinctive parts, and use these grounded parts to predict object labels. We then introduce **PARTONOMY**, a comprehensive benchmark for the *Explanatory Part Segmentation* task. We construct **PARTONOMY** from existing part segmentation datasets [11, 7, 35] and our manually-annotated evaluation dataset of 1K specialized object-centric images with complex part annotations. This subset, **PARTONOMY**-Core, contains 862 distinct part labels—more than any existing part datasets (Table 1).

We then note two shortcomings of existing segmenting LMMs' architectures [19, 37, 41]. First, they always rely on special [SEG] tokens not seen during pretraining, potentially hindering downstream performance by introducing distribution shift. Second, these segmenting LMMs discard their predictions after each output, missing the opportunity to incorporate prior information contained by the masks they predicted during the decoding process. This design is in contrast to modern generative frameworks, which condition future predictions on past ones [48, 44]. Based on these observations, we propose **PLUM**, a **P**art-**L**evel **U**nderstanding L**M**M. **PLUM** uses a text span tagging module to avoid special segmentation tokens that induce distribution shift from the pre-trained LLM, and employs a mask feedback mechanism to condition on past predictions (Section 4). Our results show that pretrained **PLUM** retains its general reasoning abilities far better than other segmenting LMMs, achieving stronger zero-shot segmentation performance and competitive finetuned performance to models trained on significantly more segmentation data.

## 2 Related Work

**Reasoning in Large Multimodal Models** Reasoning capabilities in Large Language Models (LLMs) uncovered by prompting techniques such as Chain-of-Thought (CoT) [51, 18] have led to increased interest in their application to LMMs [31, 55, 12]. Previous work shows that reasoning abilities of LLMs allow them to generate textual rationales given image inputs, allowing them to handle complex visual reasoning tasks such as A-OKVQA [42] and ScienceQA [31]. Nonetheless, LLMs' output space is confined to text, limiting their spatial understanding and often leading to hallucinatory text outputs [26]. While recent efforts on visual compositional reasoning attempt to mitigate the gap between the text and image modalities in LMMs [54, 55, 32] by using external modules such as object detectors [6] or code interpreters [46], most attempts don't truly reflect the innate visual reasoning capabilities of LMMs. Our proposed model and a recent line of LMMs

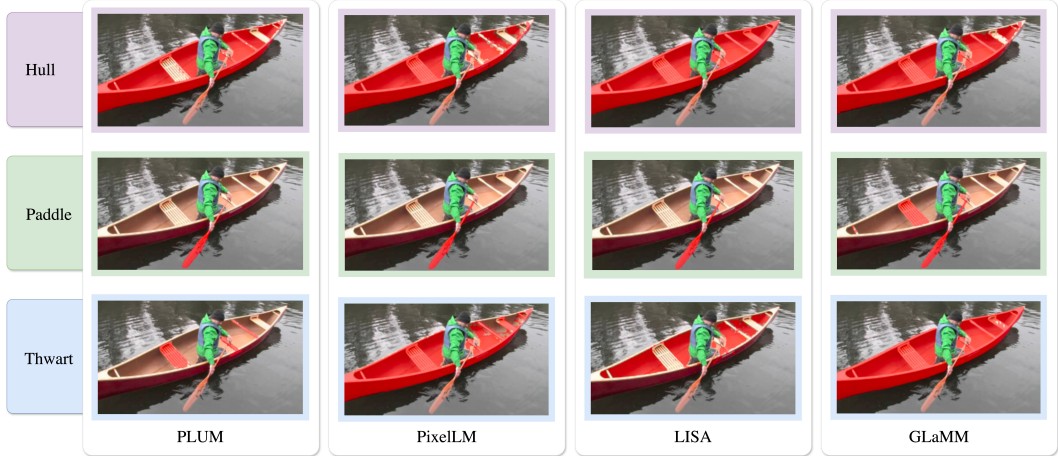

Figure 2: An example of PLUM's part understanding compared to recent segmenting LMMs trained on part data.

[19, 37, 41] try to accomplish this by interleaving textual and visual rationale generated through segmentations.

**Segmentation-Enabled Large Multimodal Models** LMMs such as LISA [19] and GLaMM [37] have demonstrated the ability to generate text and grounded segmentation masks. Despite being trained on part-level segmentation datasets such as PACO [35] and Pascal-Part [9], they struggle to exhibit a part-level understanding of visual concepts. While they demonstrate the ability to understand complex textual instructions [19, 37, 41, 53], current LMMs fail to relate concept-indicative parts to their wholes, as shown in Fig. 2. Frequently, they fail to generate the specialized segmentation token (e.g., [SEG]) added to their vocabulary, leading to no masks being generated for the parts. Even state-of-the-art LMMs that seemingly "reason" struggle to establish attributive relationships between objects and parts, implying that current models and datasets lack the coverage and capacity to handle complex part understanding and grounding tasks. This observation motivates the proposal of our new Explanatory Part Segmentation task, which requires LMMs to segment objects' parts (i.e. producing visual rationale) while generating the corresponding text rationale.

**Part Semantic Segmentation** Part segmentation is the task of decomposing objects into their constituent parts through segmentation [7, 56, 57, 11, 35]. While this task has been studied in open-vocabulary [24, 25, 59, 40] and multiple segmentation tasks [20], no existing work has evaluated LMMs' ability to segment objects' "concept-indicative" parts—those that help define the object category. In fact, most recent efforts on segmentation-enabled LMMs focus on concept labels or referring expressions [37], ignoring part segmentation altogether. Our **PARTONOMY** dataset, which includes our manually annotated **PARTONOMY**-Core evaluation set, integrates existing part-level segmentation datasets such as PACO [35] and PartImagenet [11] to further the part and object-level diversity of our benchmark.

## 3 PARTONOMY: A Dataset for Explanatory Part Segmentation

### 3.1 Task Overview

We motivate our task definition by characterizing a model with part understanding. First, such a model should be able to **identify parts**. Given an image of an object, the model can list visible parts and ground them in the image. Second, this model should be able to **compare and associate object parts**. It recognizes that both dogs and tables have legs, despite their difference in form. On the other hand, it understands that a passenger plane and a biplane both have wings, but that the biplane's double wings are a feature that distinguishes it from other aircraft. Finally, this model should be able to use its part knowledge to **predict object labels based on their parts**. Identifying a key feature, like a large scope, suggests to the model that a rifle is likely a sniper rather than assault rifle.

To evaluate these elements of part understanding, we define the **Explanatory Part Segmentation** task. In this task, a model is provided with an image and a question about an object's parts (e.g.

Table 1: **Comparison between part segmentation datasets.** † indicates usage in **PARTONOMY**. "C" refers to common objects (e.g. `chair`, `airplane`), "O" to organisms (e.g. `dogs`, `snake`), and "S" to specialized objects (e.g. `intersecting lines`, `highway map`, `fighter jet`). **PARTONOMY**-Core has over three times as many object labels and four times as many part labels as the widely used PACO dataset, has more part labels than PartImageNet++ (which has twice as many object labels), and contains specialized object parts annotated on object-centric images.

| Datasets | # Object Labels | # Part Labels | # Object-Part Labels | # Images | # Seg. Masks | Object-Centric Images | Object Domain |
|---|---|---|---|---|---|---|---|
| PASCAL-Part† | 20 | 30 | 193 | 10,103 | 111,960 | ✗ | C, O |
| PartImageNet† | 158 | 14 | 14 | 24,000 | 112,000 | ✓ | C, O |
| PartImageNet++ | 1000 | 818 | 3,308 | 100K | 406.4K | ✓ | C, O |
| PACO† | 75 | 200 | 456 | 84,027 | 641,000 | ✗ | C |
| **PARTONOMY**-Core† | 534 | 862 | 1,976 | 1,068 | 4,968 | ✓ | C, O, S |
| **PARTONOMY** | 606 | 975 | 2,507 | 74,500 | 407,101 | ✓ | C, O, S |

"What visible parts does the agricultural plane in the image have?" Fig. 1). The model must then select the best response and generate segmentation masks for the corresponding parts to explain its selection (e.g. "The agricultural plane has wings, a propulsion component, and a spraying rig"). Motivated by our characterization of part understanding, we define three classes of questions (Fig. 1):

**Part Identification** questions ask the LMM to identify then segment an object's visible parts. These questions test LMMs' ability to recognize and ground parts without hallucination.

**Part Comparison** questions ask the LMM to identify an object's visible parts and compare or contrast them to the parts of another object. These questions test models' knowledge of objects' common parts. Concretely, let $P_I$ and $P_C$ be the parts of an object in the image and the parts of a separate comparative concept. We define two subtasks:

- *Part Intersection.* The model is asked which visible parts the object in the image has in common with a specified query concept, $P_I \cap P_C$, then segments them.

- *Part Difference.* The model is asked which visible parts the object in the image has which the query concept does not, $P_I \setminus P_C$, then segments them.

**Part-Whole Reasoning** asks the LMM to identify an object or its parts as a consequence of the other. These questions assess whether the model can apply its part knowledge to identify objects, or use an object to identify its parts. Subtasks include:

- *Part-to-Whole*: The model is asked to identify and segment an object's visible parts, and based on the predicted parts, determine the object label.

- *Whole-to-Part*: The model is asked to identify the object in the image, and based on the predicted object, identify and segment its visible parts.

The subtasks assess decomposable object recognition, where an object and its parts each provide evidence for the other's identity.

## 3.2   Dataset Construction

We introduce the **PARTONOMY** dataset to facilitate training and evaluation on *Explanatory Part Segmentation*. **PARTONOMY** consists of three training and evaluation subsets—**PARTONOMY**-PACO, **PARTONOMY**-PartImageNet, and **PARTONOMY**-PASCAL Part—which are constructed from their respective datasets' part annotations [35, 11, 7]. We further contribute an evaluation-only subset of 1K images of domain-specific objects, which we term **PARTONOMY**-Core.

**PARTONOMY-Core Construction.**   To construct the **PARTONOMY**-Core ontology, we start from broad object categories containing decomposable objects—for example, *airplanes*, *garden tools*, *weapons*, and *boats* (details on dataset construction in the Appendix). We then manually select objects which provide category coverage and which have readily identifiable parts.

With object classes selected, we use the Bing search API to download a preliminary set of object images, and prompt an LLM (Llama 2-70B [47]) to generate part names for each object which are *visible* and *specific* to that object or category. We manually review each object's assigned parts,

removing those which are not outwardly visible or are not commonly found on the object. Part annotation proceeds using a combination of CVAT.AI and a mask annotation interface we developed to streamline the annotation process from multiple annotators[2]. Parts are further refined and pruned during the annotation process depending on their visibility and frequency.

**Explanatory Part Segmentation Data Generation Pipeline.**   Our *Explanatory Part Segmentation* data generation pipeline is applicable to any part dataset containing object names, part segmentations, and part names. We start with the ground truth set of object parts for each image and format these in natural language as the parts the model must identify and ground. For *Part Comparison* questions, we sample a separate object class with parts in common with those in the image, then intersect (for Part Intersection) or subtract (for Part Difference) its parts to form the ground truth set of parts.

After constructing the answer choice with the ground-truth parts, we create incorrect answer choices for each question. We adopt an *answer mutation* framework to generate plausible, challenging wrong answers. For a set of ground truth parts, we repeatedly apply mutation operations which add, remove, or replace an existing part. This process keeps wrong answers close to the original to require deep part understanding of the evaluated model. Instead of randomly sampling parts for mutation operations—which could result in unrelated part additions (e.g. "The airplane in the image has wings, a row of windows, and an ice cream cone")—we select those most related to the existing parts and object. We train logistic regressors on part co-occurrence to predict likely parts given the current set of parts, and restrict wrong answer parts to those from the same object category, if available (e.g., wrong parts for an airplane come from other airplanes' parts). Challenging wrong object answers for *Part-Whole Reasoning* questions are sampled in a similar way, selecting objects with high semantic similarity to the ground truth object as measured by word embeddings (we use Sentence Transformers [39] to measure similarity).

**Differences from Existing Datasets.**   PARTONOMY is the only part segmentation dataset designed for use with VLMs, testing not only part identification but also part reasoning and grounding. The data generation pipeline's extensibility and capacity to generate challenging questions serve as an important asset for future part-based pretraining and evaluation.

PARTONOMY-Core has, to our knowledge, the most part classes of any part segmentation dataset, with four times the number of part labels of the widely used PACO dataset [35] and more part labels than PartImageNet++ [22] which has twice as many object classes. It has object-centric images for consistent evaluation, unlike datasets like PACO, which frequently have partially occluded objects. PARTONOMY-Core is lightweight to evaluate on, with only 1K images, but covers a wide range of concepts with balanced instances (2 images per object), more than any dataset other than PartImageNet++. These qualities, coupled with PARTONOMY-Core's use of technical domains with highly-specific objects (e.g., electric coffee grinder, city map, and combat drone), make it a unique contribution for part segmentation evaluations.

## 3.3   Evaluation

Explanatory Part Segmentation requires models to choose the correct textual response and segment parts in the image.

**Text Evaluation**   To evaluate textual part predictions, we prefer a multiple choice over a generative setting to avoid ambiguities in phrasing—e.g., where the model identifies a clip but the annotations list the part as a clamp—and incomplete part annotations, e.g., where the model identifies a valid part that isn't annotated. We provide the model with one correct and four incorrect answer choices. Incorrect choices either include non-visible parts or lack visible parts present in the correct response. We select the predicted answer choice via language modeling probability, as is common in VQA [2, 21]. For Part-Whole Reasoning questions, we select an answer choice twice in sequence: once to predict the set of parts, and once to predict the object.

We evaluate answer selection via accuracy (random = 20% for 5 answer choices) for both part prediction (all questions) and object prediction (Part-Whole Reasoning questions). However, some wrong answer choices are better than others. Precision and recall capture the similarity of the predicted parts to the ground truth set, and we adopt these as more fine-grained measurements of part recognition by the LMMs.

---

[2]This interface and the dataset generation pipeline will be released to facilitate future dataset construction.

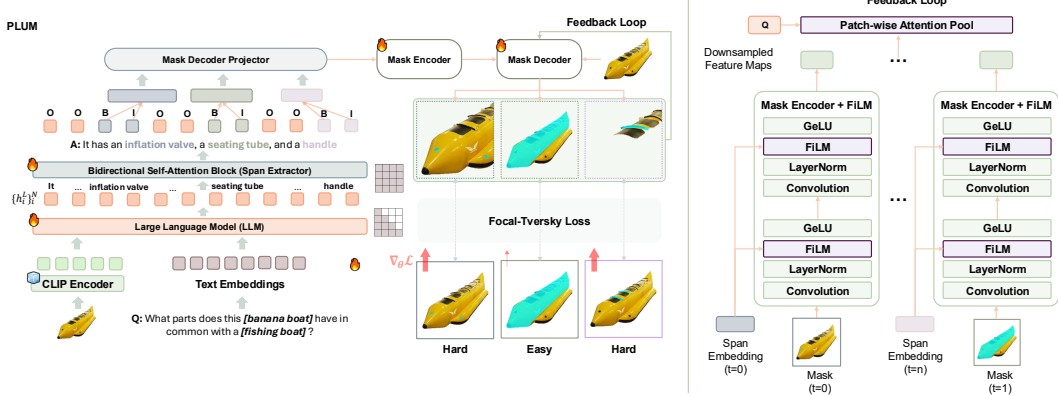

Figure 3: **Overview of PLUM. PLUM** is not dependent on special tokens (e.g., `<SEG>`) added during finetuning to generate segmentation masks. **PLUM** uses a bidirectional span extractor that automatically determines which tokens should be passed to the mask decoder to generate segmentations. A feedback loop based on SAM's mask decoder enables **PLUM** to condition future segmentations on those past.

**Segmentation Evaluation**    We evaluate part segmentations via gIoU (global IoU), which measures the average IoU over part annotations [19, 37]. *micro*-gIoU averages part IoUs over all masks in the dataset, measuring how well the model segments the most common parts. *macro*-gIoU averages part IoUs for each image, then averages these image IoUs over the entire dataset. This metric is less affected by common parts, measuring how well a model segments parts in general.

## 4    PLUM: Part-Level Understanding LMM

**Shortcomings of LMMs on Part Understanding**    We find that existing LMMs are unable to accurately identify parts in an image. Even segmenting LMMs [19, 37, 41] trained on part segmentation datasets such as PACO [35] and Pascal-Part [7] exhibit poor performance on part-level segmentation (Fig. 4 and Table 2). We identify two key architectural deficiencies of segmenting LMMs: (1) They rely on special tokens for segmentation (e.g. [SEG] or `<p></p>`). These tokens are not seen during pretraining, so we hypothesize that their addition to the vocabulary and subsequent finetuning perturbs models' original token distributions (Table 5). (2) They discard prior mask predictions when segmenting in sequence, conditioning only on past text during generation. Incorporating prior mask predictions would likely help maintain consistency and better localize future predictions (Fig. 5a).

**Proposed Method**    Based on these observations we propose **PLUM**, a segmenting LMM with part-level understanding. **PLUM** consists of a vision-language model (initialized from LLaVA [28]) which takes image and text inputs, along with a mask decoder (initialized from SAM's decoder [17]) that generates segmentation masks.

Let $h_i^L \in \mathbb{R}^d$ be the VLM's last-layer embedding of token $i$ $(i = 1, \ldots, N)$ of the output sequence. We process these embeddings along two complementary pathways: (i) the **Span Extractor**, a bidirectional self-attention block that tags beginning ($B$), inside ($I$), and outside ($O$) [36] positions of tokens to segment, and (ii) a projection head that maps $B/I$ embeddings into "mask queries," regularized by KL divergence. An overview is given in Fig. 3.

**Token-level Query Selection (*Span Extractor*)**    The *Span Extractor* enables the selection of segmentation-relevant text spans to pass to the mask decoder without the use of a dedicated segmentation token. Given the last-layer token embeddings $\{h_i^L\}_{i=1}^N$, we apply a two-layer token-wise MLP before infusing global context by passing the embeddings through a bidirectional Transformer encoder block. This bidirectional attention is critical for reliable BIO span tagging, as otherwise the LLM's causal masking prevents embeddings from seeing future context. A final projection layer maps these contextualized embeddings to $\{B, I, O\}$ logits. We train the span extraction module using cross entropy loss $\mathcal{L}_{\text{span}}$ where $B, I$ tags correspond to part names to segment.

During inference, contiguous $B \rightarrow I$ chains are greedily merged to form text spans that are projected into segmentation queries. Note, we also enable users of **PLUM** to override the automatic tags with manual span selection, enabling interactive, interpretable "highlight-to-segment" behavior.

**Query Projection with KL Constraint**   Let $\mathcal{S} = \{(i_s, j_s)\}_{s=1}^{N_+}$ be the set of contiguous $B \rightarrow I$ spans produced by the span extractor, where $i_s$ and $j_s$ are the start and end token indices of span $s$, and $N_+ = |\mathcal{S}|$ is the total number of such spans in the sequence. For every span token $k \in [i_s, j_s]$ we obtain a "mask-query" vector $q_k = g(h_k^L) \in \mathbb{R}^m$, with $g(\cdot)$ a learned MLP projection. To keep the span representations close to the pre-trained backbone VLM's manifold, we pool the last-layer embeddings of each span[3] and impose a Gaussian KL penalty against the corresponding frozen teacher embedding $t_{i_s:j_s}^L$: $\mathcal{L}_{\mathrm{KL}} = \frac{1}{N_+} \sum_{s=1}^{N_+} \frac{\left\| h_{i_s:j_s}^L - t_{i_s:j_s}^L \right\|_2^2}{2\sigma^2}$. This term is applied only to *B/I* spans, preventing their hidden states from drifting away from the original language-representation space and thereby preserving the VLM's textual reasoning ability.

**Mask Feedback Loop**   To incorporate previously predicted masks into the mask decoding process, we inject feature-wise linear modulation (FiLM) [34] layers into the SAM decoder's mask encoder (Fig. 3). These layers allow us to encode the mask while conditioning on prior text spans, providing semantics beyond a raw binary mask. We use this modified mask encoder to encode each prior mask into a stack of text-enhanced feature maps. The stack of feature maps is pooled into a single feature map via patch-wise attention pooling (over the stack dimension), with a learned feature map providing an attentional query for each patch. This pooled feature map representing all prior predicted masks is fed into the mask decoder (along with the pooled text embeddings) to generate the next mask.

**Segmentation Mask Generation**   Tagged token embeddings $q_k$ are average pooled and passed to the mask decoder, generating a mask $\hat{M}_i$. With ground-truth mask $M_i$ we adopt the Focal-Tversky loss [1], $\mathcal{L}_{\mathrm{seg}} = \frac{1}{N_+} \sum_{y_i \neq O} \mathcal{L}_{\mathrm{FT}}(M_i, \hat{M}_i)$. Focal-Tversky loss is a generalized version of the DICE loss [45]. This gives the overall objective equal to $\mathcal{L} = \mathcal{L}_{\mathrm{LM}} + \lambda_1 \mathcal{L}_{\mathrm{span}} + \lambda_2 \mathcal{L}_{\mathrm{KL}} + \lambda_3 \mathcal{L}_{\mathrm{seg}} + \lambda_4 \mathcal{L}_{\mathrm{BCE}}$, where $\mathcal{L}_{\mathrm{LM}}$ is the standard language-generation loss and $\mathcal{L}_{BCE}$ is per-pixel binary cross-entropy as adopted from [19]. The BIO head precisely extracts segmentation spans, while the Focal-Tversky loss, biased toward recall ($\alpha = 0.7$) and precision ($\beta = 0.3$), encourages sharper, high-IoU masks at fine-grained image regions. For additional details on the hyperparameter setting, refer to §A.1

## 5   Experiments

**Implementation Details**   We use a pre-trained LMM, LLaVA-7B, and LLaVA-llama2-13B [28] as backbones for PLUM (Sec. 4). PLUM follows the consecutive two-stage finetuning process: (1) PLUM is first finetuned with a randomly sampled mixture of PACO-LVIS [35], Pascal Parts [7], PartImageNet [11], COCO-Stuff [5], ADE20k [56], the RefCOCO line of datasets [15], a VQA dataset from LLaVA (`llava_instruct_150k`), and a Reasoning Segmentation [19] dataset. This setting is similar to the previous line of segmentation-enabled LMMs [19, 37, 41], and we refer to the stage-1 checkpoint of PLUM as the zero-shot (or pretrained) baseline throughout this paper. (2) To further finetune PLUM on our **PARTONOMY** training dataset, we take **PARTONOMY**-PACO, -PartImageNet and -PascalParts to construct a training split and a validation split. Note, we do not use the **PARTONOMY**-Core split as training data and use it only as evaluation data. We refer to the Appendix for additional details on the hyperparameter settings and training details.

**Baselines**   To evaluate PLUM's proposed changes, we use LISA [19], GLaMM [37], and PixelLM [41] as our primary baselines. All of these models use LLaVA as the base LMM and use SAM-style decoders to generate segmentation masks [28, 17].

For segmentation, we also evaluate X-Decoder, SEEM, and Grounded SAM 2 as general open-vocabulary segmentation models [59, 60, 40, 38]. As they do not understand question-based prompts, we provide them with the ground truth (gt) parts to segment individually. We also include SegLLM, a segmenting LMM based on LLaVAv1.5 and HIPIE [27, 50]. HIPIE is a decoder built for multi-scale and part segmentation, and SegLLM is trained to perform multi-round segmentation on parts, allowing it to refer back to previously predicted masks. The similarity of this mechanism to our feedback loop motivates us to include SegLLM as a baseline, despite the imperfect comparison due to its use of a newer LMM and different mask decoder.

---

[3] We use mean pooling; any differentiable pooling operator is admissible.

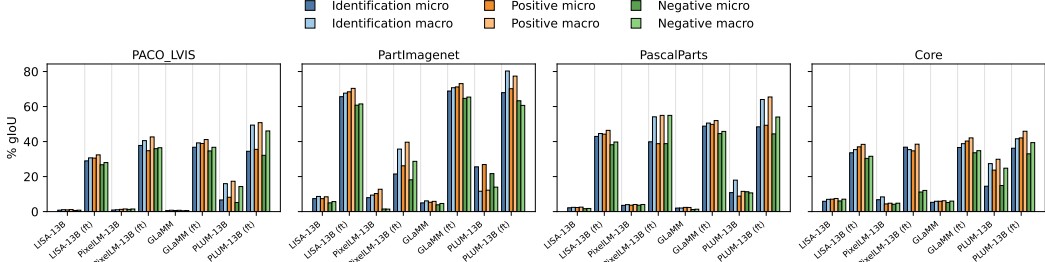

Figure 4: Performance (micro/macro gIoU) on **PARTONOMY** validation splits.

For text evaluations, we include a *random* baseline (which randomly selects answers) to situate the models' part precision and recall. GPT-4o[4], a frontier model, provides an upper bound on performance. GPT-4o has an advantage as it must be provided with all four answer choices at once, allowing it to take advantage of shortcuts not available to the other models (like identifying the base answer from which the wrong answer choices are generated).

## 5.1 Explanatory Part Segmentation

**Part Identification and Comparison Questions** In Table 2, PLUM outperforms LISA and GLaMM on all three part-segmentation question types in the zero-shot setting. We attribute this gain to (i) span-level constraints that keep pre-trained textual semantics intact and (ii) our mask-feedback loop, which refines each mask using its visual history. By contrast, Table 3 shows only marginal gaps in text-only metrics (P, R, Acc.). **PARTONOMY**'s answer choices intentionally contain extensive lexical overlap, demonstrating the language models' difficulties in comparing similar answer choices.

Table 2: **Explanatory Part Segmentation's segmentation performance (gIoU) on PARTONOMY-Core.** "ft" = fine-tuned on Partonomy; "gt" = OV segmentation models given ground-truth answers. *Part2Whole* and *Whole2Part* scores are reported only for part prediction.

| Methods | Extra Seg Data | Identification | | Intersection | | Difference | | Part2Whole | | Whole2Part | |
|---|---|---|---|---|---|---|---|---|---|---|---|
| | | micro | macro | micro | macro | micro | macro | micro | macro | micro | macro |
| *Open-Vocabulary Segmentation Models* | | | | | | | | | | | |
| X-Decoder (gt) [59] | – | 11.5 | 13.4 | 13.5 | 14.0 | 11.1 | 12.4 | – | – | – | – |
| SEEM (gt) [60] | – | 13.5 | 15.5 | 17.1 | 18.2 | 12.5 | 13.8 | – | – | – | – |
| Grounded SAM 2 (gt) [40] | – | **13.6** | **16.8** | 20.6 | 23.6 | 14.3 | 17.1 | – | – | – | – |
| *LLaVAv1.5 + HIPIE [27, 50]* | | | | | | | | | | | |
| SegLLM | – | 29.6 | 32.4 | 32.2 | 33.8 | 28.5 | 30.7 | 29.4 | 32.3 | 29.3 | 32.2 |
| *LLaVA + SAM-style Mask Decoder [28, 17]* | | | | | | | | | | | |
| LISA-13B [19] | ✗ | 5.9 | 7.0 | 7.1 | 7.5 | 6.1 | 7.1 | 5.7 | 6.6 | 6.0 | 6.8 |
| PixelLM-13B [41] | ✓ | 6.8 | 8.4 | 4.4 | 4.8 | 4.2 | 4.8 | 4.6 | 5.4 | 6.3 | 7.8 |
| GLaMM [37] | ✓ | 5.3 | 5.9 | 5.9 | 6.2 | 5.2 | 6.0 | 4.8 | 5.6 | 4.9 | 5.8 |
| **PLUM-13B** | ✗ | **14.5** | **27.4** | **23.7** | **29.9** | 14.9 | 24.8 | **14.3** | **26.8** | **15.4** | **27.5** |
| LISA-13B (ft) | ✗ | 33.6 | 35.4 | 37.0 | 38.4 | 30.4 | 31.6 | 32.6 | 34.7 | 34.3 | 36.2 |
| PixelLM-13B (ft) | ✓ | **36.8** | 35.4 | 34.7 | 38.5 | 11.2 | 12.1 | 34.9 | 33.6 | 32.9 | 34.2 |
| GLaMM (ft) | ✓ | 36.6 | 38.8 | 40.3 | 42.1 | **33.6** | 34.8 | 36.1 | 38.5 | 35.7 | 38.0 |
| **PLUM-13B (ft)** | ✗ | 36.2 | **41.6** | **42.1** | **45.9** | 33.0 | **39.4** | **36.7** | **40.8** | **36.2** | **39.8** |

**Part-Whole Reasoning Questions** The Part-Whole Reasoning results of Table 2 show that knowing the object label prior to part segmentation leads to better mask prediction performance—the pretrained models obtain higher Whole2Part than Part2Whole scores. This suggests that part mask prediction benefits from object label conditioning. The advantage obtained by object conditioning evaporates once the models have been trained on sufficient part data, however, as shown by the finetuned models. Similarly, in Table 3 the models' increase in object accuracy after conditioning on the object's parts, and their increase in part accuracy after conditioning on the object label, underscores the utility of jointly predicting object labels and parts.

---

[4]Specifically, `gpt-4o-2024-08-06`.

Table 3: **Explanatory Part Segmentation text performance on PARTONOMY-Core.** "ft" = fine-tuned on the Partonomy training sets. *P* and *R* denote Precision and Recall; *A* is multiple-choice accuracy. *OA* and *PA* refer to object and part accuracy for *Part2Whole* and *Whole2Part* questions.

| Methods | Identification | | | Intersection | | | Difference | | | Part2Whole | | | | Whole2Part | | | |
|---|---|---|---|---|---|---|---|---|---|---|---|---|---|---|---|---|---|
| | *P* | *R* | *A* | *P* | *R* | *A* | *P* | *R* | *A* | *P* | *R* | *PA* | *OA* | *P* | *R* | *PA* | *OA* |
| LISA-13B [19] | 88.4 | 67.3 | 21.9 | **68.5** | 57.8 | **47.9** | 83.2 | 61.9 | 24.5 | 87.2 | 68.8 | 25.0 | 65.4 | 88.5 | 68.8 | 27.2 | 58.0 |
| PixelLM-13B [41] | **90.4** | 73.6 | 35.0 | 66.1 | 57.0 | 46.1 | **83.3** | 71.1 | **35.8** | **87.6** | 73.6 | 33.2 | 60.4 | **94.5** | 84.3 | 57.9 | 41.2 |
| GLaMM [37] | 83.8 | 89.5 | 48.6 | 51.5 | 69.1 | 32.3 | 74.0 | 81.8 | 32.9 | 72.8 | 93.0 | 20.1 | 62.6 | 74.1 | 93.2 | 24.1 | 50.7 |
| **PLUM** | 86.6 | **95.6** | **60.2** | 49.1 | **72.7** | 26.6 | 73.9 | **88.8** | 35.0 | 85.6 | **94.5** | **58.6** | **71.5** | 90.6 | **95.9** | **70.9** | **59.3** |
| LISA-13B (ft) | 75.5 | **98.0** | 30.7 | **71.8** | **98.2** | 35.7 | 61.0 | **96.5** | 27.5 | 73.1 | **96.5** | 23.8 | **68.8** | 74.6 | **97.3** | 28.0 | **61.3** |
| PixelLM-13B (ft) | **87.2** | 64.3 | 16.5 | 58.3 | 49.6 | **36.9** | **83.3** | 60.8 | 22.7 | **84.8** | 64.2 | 16.1 | 49.9 | **87.2** | 66.3 | 22.1 | 36.0 |
| GLaMM (ft) | 75.9 | 87.2 | 30.1 | 48.1 | 68.1 | 26.0 | 68.6 | 81.1 | 21.9 | 80.4 | 78.5 | 31.8 | 56.3 | 79.6 | 78.3 | 32.0 | 43.6 |
| **PLUM (ft)** | 82.9 | 85.0 | **42.1** | 53.6 | 69.9 | 34.7 | 73.3 | 77.8 | **30.2** | 81.7 | 84.0 | **40.9** | 60.0 | 84.8 | 88.1 | **46.3** | 51.2 |
| Random | 76.2 | 76.7 | 20.0 | 44.1 | 54.1 | 20.0 | 70.9 | 73.3 | 20.0 | 75.3 | 76.7 | 20.0 | 20.0 | 75.5 | 76.5 | 20.0 | 20.0 |
| SegLLM | 90.6 | 86.4 | 51.9 | 61.7 | 70.2 | 47.0 | 77.1 | 79.8 | 73.3 | 92.3 | 81.1 | 48.4 | 69.7 | 94.2 | 89.3 | 65.3 | 62.8 |
| GPT-4o | 95.6 | 96.6 | 81.7 | 78.9 | 84.9 | 70.7 | 92.1 | 92.4 | 72.3 | 97.7 | 97.9 | 89.0 | 96.5 | 97.7 | 97.9 | 89.2 | 96.3 |

## 5.2 Additional Downstream Tasks and Ablation Study

We further evaluate **PLUM** on non part-centric downstream tasks to evaluate its generalizability and whether our choice to omit special [SEG] tokens preserves **PLUM**'s pretraining knowledge. For segmentation, we choose the Reasoning Segmentation task [19], which requires the model to reason about the referenced object before segmenting it. To evaluate **PLUM**'s general visual reasoning, we select the Visual Question Answering (VQA) tasks TextVQA [43] and GQA [13], and a visual hallucination task, POPE [23].

**Reasoning Segmentation** **PLUM** demonstrates strong generalization to the Reasoning Segmentation task proposed in [19]. As shown in Table 4, **PLUM** outperforms existing open-vocabulary segmentation models such as X-Decoder and OVSeg, and also generates more accurate segmentations than LISA, the model proposed for the task.

Table 4: **Performance on the ReasonSeg benchmark.** **PLUM** outperforms open-vocabulary segmentation models and LISA, the original model trained for reasoning segmentation.

| Method | gIoU | Method | gIoU |
|---|---|---|---|
| OVSeg [24] | 28.5 | LISA-7B [19] | 44.4 |
| X-Decoder [59] | 22.6 | LISA-7B (ft) | 52.9 |
| SEEM [60] | 25.5 | LISA-13B | 48.9 |
| Grounded-SAM [40] | 26.0 | LISA-13B (ft) | 56.2 |
| LLaVA1.5-7B + OVSeg | 38.2 | **PLUM-7B** (ft) | **53.5** |
| LLaVA1.5-13B + OVSeg | 37.9 | **PLUM-13B** (ft) | **57.3** |

Table 5: Accuracy (%) on VQA datasets and the POPE hallucination benchmark. Numbers in parentheses show percentage change relative to the LLaVA-13B backbone. LISA and PixelLM lose most of their vision-language reasoning abilities upon finetuning for segmentation, whereas **PLUM**'s performance increases.

| Method | TextVQA | GQA | POPE |
|---|---|---|---|
| LISA-13B [19] | 1.58 (−93.1%) | 1.33 (−97.4%) | 1.87 (−94.1%) |
| PixelLM-13B [41] | 8.37 (−63.4%) | 21.72 (−57.6%) | 15.29 (−51.9%) |
| LLaVA-13B [28] | 22.84 | **51.27** | 31.80 |
| **PLUM-13B (ours)** | **30.11** (+31.8%) | 39.18 (−23.6%) | **34.65** (+8.9%) |

**Distribution Shift Induced by Special Tokens** To investigate whether **PLUM**'s removal of special tokens mitigates distribution shift from LLaVA's pre-trained representations, we compare **PLUM**'s performance on VQA [43, 13, 23] tasks to those of models which use [SEG] tokens and to that of the base LLaVA model Table 5. To our surprise, the performance of state-of-the-art segmenting LMMs deteriorates significantly on VQA and visual hallucination [23] tasks. Through inspection, we find that LISA tends to only generates irrelevant [SEG] tokens. While PixelLM performs somewhat better, it still suffers due to its use of multiple specialized tokens [41]. In contrast, **PLUM** outperforms the other segmenting LMMs, even outperforming the LLaVA backbone on 2/3 tasks. This strong

performance demonstrates that our paradigm of segmentation through span extraction preserves the visual reasoning capacity of the LMM.

Figure 5: **Ablations** (a) The feedback loop and tagging mechanism improve part segmentation on **PARTONOMY**-PartImageNet; (b) Varying the KL-constraint weight $\lambda_{KL}$ trades off segmentation gIoU and TextVQA accuracy.

(a) Ablating key components of PLUM. $F$ stands for the feedback loop. LISA has neither feedback loop nor span extractor.

| Model(s) | micro-gIoU | macro-gIoU |
|----------|------------|------------|
| LISA-13B | 65.6 | 67.7 |
| LISA-13B $(+F)$ | 66.7 | 68.6 |
| PLUM-13B $(-F)$ | 61.4 | 73.9 |
| PLUM-13B | **67.9** | **80.3** |

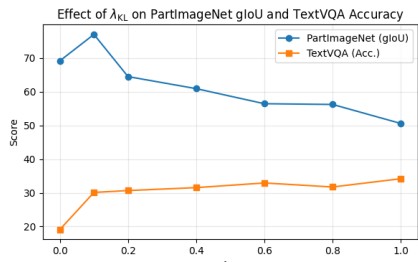

(b) Effect of $\lambda_{KL}$ on segmentation (PartImageNet) and visual reasoning (TextVQA).

**Ablating Key PLUM Components**  Removing the iterative mask-feedback loop lowers micro-gIoU by 9.6% and macro-gIoU by 8%, though the span-based tag extractor alone still beats the LISA-13B baseline by 8.4% macro-gIoU. When both components are active, **PLUM** tops LISA by 3.5% (micro) and 20% (macro), showing that tag extraction broadens long-tail coverage while feedback refines segmentation accuracy.

**Effect of KL Divergence**  Increasing the KL-alignment weight $\lambda_{KL}$ from 0 to 1.0 steadily trades segmentation for reasoning: PartImageNet micro-gIoU falls by nearly 20%, whereas TextVQA accuracy sees a 75% performance improvement. We set $\lambda_{KL} = 0.1$ in this work.

## 6  Limitations and Broader Impacts

**Limitations**  Our work advances fine-grained, part-level understanding but still faces several constraints. Although **PARTONOMY**-Core includes the largest number of part labels to date, with 862 categories across 534 objects, it omits some rare or domain-specific parts and lacks the object diversity of [11]. Expanding its coverage would further enhance LMM comprehension. While **PLUM** mitigates major architectural limitations of prior segmenting LMMs via BIO tagging and a FiLM-based feedback loop, it still struggles with small or ambiguous parts and may not scale efficiently to high-resolution imagery.

**Broader Impacts**  Part grounding is crucial for compositional visual reasoning and safety-critical domains such as robotic manipulation, assistive systems, and automated inspection. Such scenarios require accurate and interpretable part understanding, with mistakes in identifying and the grounding of parts leading to catastrophic consequences. By introducing **PARTONOMY** and **PLUM**, we aim to foster more reliable and interpretable LMMs. The high computational cost of large segmenting LMMs also underscores the need for more efficient, sustainable architectures. We hope our benchmark and analyses inspire continued research toward robust, efficient, and responsible part-grounding methods.

## 7  Conclusion

We introduce *Explanatory Part Segmentation* with the **PARTONOMY** benchmark to evaluate part-level visual reasoning and segmentation at scale. **PARTONOMY** spans 606 object labels and 2,507 part labels; its evaluation split, **PARTONOMY**-Core, alone contributes 534 objects, 862 unique parts, 1,068 images and 4,968 pixel masks—tripling PACO's object diversity and quadrupling its part vocabulary. By analysing current segmentation-enabled LMMs, we pinpoint two systemic flaws—(i) distribution-shift from pre-trained weights and (ii) discarded visual context—and address them with **PLUM**, a span-tagging, mask-feedback LMM that interleaves textual and visual reasoning without extra tokens. **PLUM** outperforms fine-tuned LISA-13B on ReasonSeg and adds 31.8% relative performance improvement on TextVQA compared to the LLaVA-13B backbone. Together, **PARTONOMY** and **PLUM** lay a quantitative and methodological foundation for future research on fine-grained, compositional, and interpretable multimodal models.

# 8 Acknowledgements

This research is based upon work supported by U.S. DARPA ECOLE Program No. #HR00112390060. The views and conclusions contained herein are those of the authors and should not be interpreted as necessarily representing the official policies, either expressed or implied, of DARPA, or the U.S. Government. The U.S. Government is authorized to reproduce and distribute reprints for governmental purposes notwithstanding any copyright annotation therein.

This work used Delta at the National Center for Supercomputing Applications through allocation #250183 from the Advanced Cyberinfrastructure Coordination Ecosystem: Services & Support (ACCESS) program [4], which is supported by U.S. National Science Foundation grants #2138259, #2138286, #2138307, #2137603, and #2138296.

The authors thank Steven Gomez and MIT Lincoln Laboratory for proposing an initial set of part-centric concepts as part of the DARPA ECOLE Program. We also thank the numerous annotators who contributed to the Partonomy dataset.

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

# A Appendix

## A.1 Model Setup

| Hyperparameter | PLUM | LISA | PixelLM | GLaMM |
|---|---|---|---|---|
| *Backbone* | | | | |
|   Language model | LLaMA-2-13B | LLaMA-2-13B | LLaMA-2-13B | LLaMA-2-13B |
|   Vision tower | CLIP ViT-L/14 | CLIP ViT-L/14 | CLIP ViT-L/14 | CLIP ViT-L/14 |
|   Mask decoder | SAM ViT-H | SAM ViT-H | Conv-U-Net | SAM ViT-H |
| *I/O and training schedule* | | | | |
| Input resolution (px$^2$) | $1024^2$ | $1024^2$ | $1024^2$ | $1024^2$ |
| Max text length | 512 | 512 | 512 | 512 |
| Precision | bf16 | bf16 | bf16 | bf16 |
| Epochs | 25 (0-shot) \| 4 (ft) | / \| 4 (ft) | / \| 4 (ft) | / \| 4 (ft) |
| Batch size | 6 | 6 | 6 | 6 |
| Grad. accumulation | 10 | 10 | 10 | 10 |
| Effective batch | 10× bsz × GPU | 10× bsz × GPU | 10× bsz × GPU | 10× bsz × GPU |
| *Optimizer* | | | | |
| Optimizer | AdamW | AdamW | AdamW | AdamW |
| Learning rate | $3\times10^{-4}$ | $3\times10^{-4}$ | $3\times10^{-4}$ | $3\times10^{-4}$ |
| Betas | (0.9, 0.95) | id. | id. | id. |
| Weight decay | 0 | 0 | 0 | 0 |
| Gradient clip | 1.0 | 1.0 | 1.0 | 1.0 |
| *Loss weights* | | | | |
| $\lambda_{CE}$ (LM) | 1.0 | 1.0 | 1.0 | 1.0 |
| $\lambda_{seg}$ (Dice/FTL) | 8.0 (FTL$^\dagger$) | 0.5 (Dice) | 0.5 (Dice) | 0.5 (Dice) |
| $\lambda_{BCE}$ (mask) | 2.0 | 2.0 | 2.0 | 2.0 |
| $\lambda_{KL}$ | 0.1 | — | — | — |
| $\lambda_{cls}$ (BIO) | 2.0 | — | — | — |
| Dice scale factor | $10^3$ | $10^3$ | $10^3$ | $10^3$ |
| FTL $(\alpha, \beta)$ | (0.7, 0.3) | — | — | — |
| *Additional Modules* | | | | |
| BIO span tagger | ✓ | — | — | — |
| Bidirectional encoder | 2048 | — | — | — |
| Feedback Loop (*Temporal Mask Pooler*)) | ✓ | — | — | — |
| Trainable SAM parts | decoder + prompt-enc. | decoder | — | decoder |
| LoRA on LM (q,v) | $r$=8, $\alpha$=16, $p$=0.05 | id. | id. | id. |

Table 6: **Hyperparameters used for all experiments.** We juxtapose four segmenting LMMs, including PLUM, against each other to illustrate the hyperparameter differences among the models. "id." indicates "identical" to the other model's setting, "—" indicates "not applicable", and "bsz" indicates the batch size the models are trained on. $^\dagger$PLUM defaults to Focal-Tversky loss; when we ablate it we fall back to standard Dice. $^\ddagger$Cross-attention is enabled only in ablation runs when explicitly specified.

**Model Training and Evaluation**   PLUM is optimized in two stages: *Stage-0* ("0-shot") mixes four publicly-available multi-task corpora—semantic segmentation [35, 11, 7, 56], referring segmentation [15], visual-question answering and image captioning [28] (9:5:5:1 sampling ratio)—for 25 epochs. *Stage-1* then optionally finetunes on **PARTONOMY**-PACO, **PARTONOMY**-PartImageNet, **PARTONOMY**-PascalPart training splits for an additional 4 epochs, and we call this PLUM checkpoint **PLUM (ft)** as shown in Tables 2 and 3. We resize every image to $1024^2$ pixels and truncate text to 512 tokens.

Training uses DeepSpeed ZeRO-2 with `bf16` precision, a per-GPU batch of 6, and `gradient_accumulation_steps` = 10 (effective batch $10 \times$ bsz $\times N_{GPU}$). Weights are updated by AdamW ($\beta$=(0.9, 0.95), no weight-decay) with a peak learning-rate of $3 \times 10^{-4}$, linearly warmed up for the first 100 optimization steps and clipped to a global norm of 1.0 thereafter.

We freeze the CLIP vision tower and the MM projection layer; all other components are trainable. The LLaMA decoder receives LoRA adapters on `q_proj`,`v_proj` ($r$=8, $\alpha$=16, dropout 0.05). On the vision side we finetune the SAM mask-decoder and, when specified, the prompt-encoder. PLUM's additional heads, the bidirectional BIO encoder and the Temporal Mask Pooler for mask feedback loop, are always optimized.

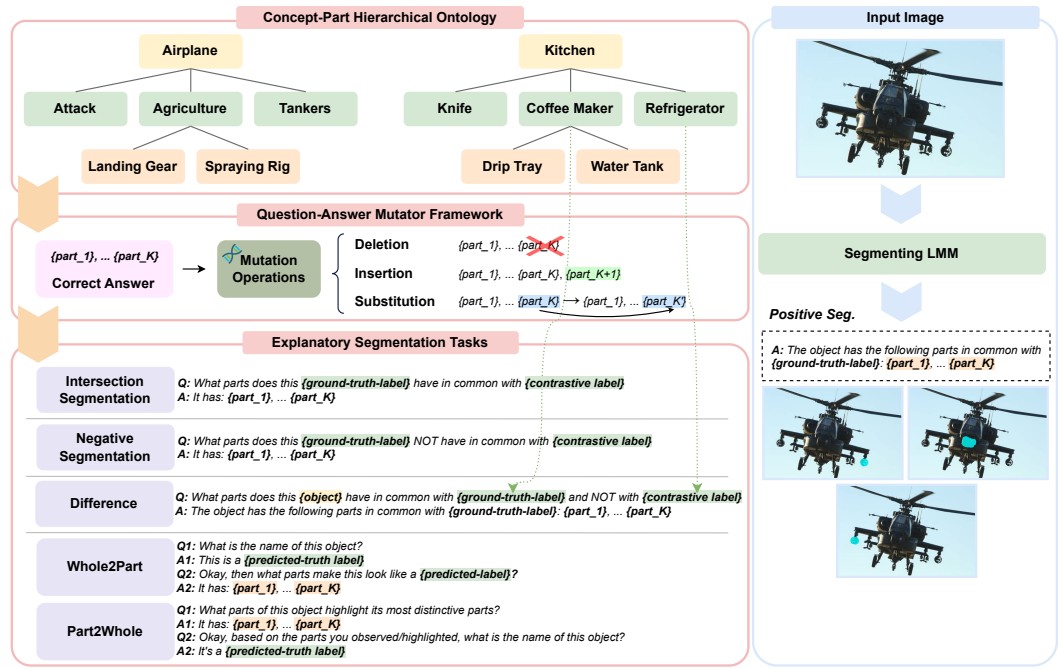

Figure 6: Illustration of the hierarchical object-part ontology construction in **PARTONOMY**-Core. We manually collect 562 object-level concepts and generate part-level concepts using LLM and manually filter out overlapping parts. Our question-answer mutator then generates challenging answer choices based on the part set overlap between object-level concepts.

The total loss is a weighted sum of (i) language modeling cross-entropy, (ii) BIO span classification loss, (iii) Focal-Tversky[5] and pixel-wise BCE for masks, (iv) KL divergence to a frozen teacher snapshot of LLaMA. Loss weights follow Table 6; random seed is fixed to 42.

For model evaluation, we first divide the performance evaluation to two facets: (i) pixel-level mask prediction evaluation as in Table 2, and (ii) multiple choice answer selection evaluation. Note, for multiple choice answer selection, we take the argmin over the entropy of each answer choice (i.e., the argmax over the sequence-level probability of each answer choice), and greedily select the answer choice with the lowest entropy. For pixel-level mask prediction, we provide the ground-truth answer choice and their part text (or [SEG] for LISA, PixelLM and GLaMM) so that the models can be evaluated solely on their mask prediction performance per part text.

# B  PARTONOMY Construction

**Ontology construction**  For **PARTONOMY**-Core, we first construct the object-part hierarchy as illustrated in Figure 6. The hierarchy starts with a set of root-level, superordinate object categories (e.g., airplane, kitchen tools), where each category contains a set of intermediate object-level concepts (e.g., agricultural airplane, coffee maker) that fall under each superordinate category, and part-level concepts which compose the leaf nodes of each object-level concept. We manually define 10 distinct superordinate concept categories, ranging from generic concepts, e.g., vehicle, office supplies, to more complex concepts such as geography[6]. There are a total of 534 object-level concepts (e.g., banana boat under the boats category) These concepts contain 1976 concept-specific parts (e.g. biplane-wing, or 862 unique part types, each appearing in 1.6 object categories on average. Table 7 shows the full list of object categories with example object labels.

---

[5]Dice loss when focal_tversky=False.

[6]We thank MIT Lincoln Lab and DARPA for developing an initial category and concept ontology as part of the DARPA ECOLE effort, which we refine and expand to construct our part-centric ontology.

| Object Category | Object Label Examples |
|---|---|
| airplanes | *agricultural, fighter jet, ultralight* |
| boats | *amphibious, barge, submarine* |
| drones | *firefighting, combat, recreational* |
| garden | *hose, hand rake, hedge shears* |
| geography | *airport, hot spring, roadmap* |
| geometry | *angle, intersecting lines, ray* |
| helicopter | *attack, medevac, law enforcement* |
| kitchen | *air fryer, dish brush, soup spoon* |
| office supplies | *shredder, staple remover, legal envelope* |
| ships | *aircraft carrier, corvette, oil tanker* |
| tools | *adjustable wrench, level, vise* |
| vehicles | *bulldozer, racing car, tricycle* |
| weapons | *anti-ballistic missile, handgun, sword* |

Table 7: List of object categories with example object labels from each.

Table 8: **Full PARTONOMY statistics after normalizing object and part names across subsets.** For PartImageNet we define an "object" as its category (containing multiple classes), as the parts are the same across each category. There are more object and part categories in the validation set as **PARTONOMY**-Core is evaluation only.

| Datasets | # Object Labels | # Part Labels | # Object-Part Labels | # Images | # Seg. Masks |
|---|---|---|---|---|---|
| **PARTONOMY** (train) | 89 | 216 | 563 | 58,706 | 321,751 |
| **PARTONOMY** (val) | 606 | 975 | 2,493 | 15,794 | 85,350 |
| **PARTONOMY** | 606 | 975 | 2507 | 74,500 | 407,101 |

**Image Annotation**  Our image annotation consists of two stages: (i) Mask annotation with the first group of annotators; (ii) Mask re-annotation and revision with the second group of annotators (the authors of this paper) to ensure the quality and correctness of the mask annotation and the parts associate with each coarse-level concept category. For the first stage of our mask annotation, we first crawl the images using the object-level and part-level concept names in our ontology hierarchy as queries and manually annotate masks for each image with human annotators. We assign each annotator with at least 10 coarse-level concept categories and provide instructions to use an external image annotation tool[7] for the segmentation mask annotation. For the parts with fewer than $m$ annotations across the dataset, we remove the parts associated with a concept; we set a hard threshold at $m = 5$.

**Building off of Existing Part Segmentation Datasets**  Our pipeline allows us to generate questions-answer pairs from any set of existing part annotations. Therefore, we use the Pascal-Part [7], PartImageNet [11], and PACO [35] datasets to expand the diversity of our task data. We take the training splits from each, divide them into training and evaluation sets by images in an 80/20 ratio, then generate at most one question of each type for each image. Depending on the question type, number of objects in the image, and the objects' parts, generating multiple questions for a single image is possible—however, to maintain dataset balance we cap the number of questions per question type for each image to one.

The PACO dataset frequently has multiple instances of the same object class in an image. To disambiguate the referenced object when asked to ground parts of "the object", we annotate the images with bounding boxes for those with multiple object instances. We select the object instance which is largest and which has the most annotated parts for use in our dataset.

Dataset statistics for our merged **PARTONOMY** dataset, containing all subsets (Pascal Part, PartImageNet, PACO, and **PARTONOMY**-Core) can be found in Table 8.

---

[7]https://www.cvat.ai/

# C Additional Experiments

To examine how Explanatory Part Segmentation models behave beyond the Partonomy Core split, we evaluate the same set of systems on three public, large–scale part–segmentation benchmarks that vary in object vocabulary size and annotation granularity: PACO_LVIS [35], PartImageNet [11], and PascalParts [7]. The quantitative results are summarized in Tables 9–11. Below we highlight the key findings.

## C.1 Zero-shot generalization

In Tables 9, 10 and 11, we show that PLUM-13B generalizes best without extra segmentation data. Despite using no part masks during pre-training, PLUM-13B outperforms every other zero-shot model on all three datasets (e.g., Identification macro-gIoU = 16.0 on PACO_LVIS vs. $\leq 1.1$ for baselines; Difference macro-gIoU = 14.3 vs. $\leq 0.8$). The gains are most pronounced on PACO_LVIS—an open-vocabulary benchmark with 406 object categories—suggesting that PLUM's part mask-language alignment carries over to out-of-distribution objects with minimal degradation.

Additional segmentation-supervision helps but is not sufficient. PixelLM-13B and GLaMM leverage large-scale mask supervision during pre-training (✓in the Extra Seg Data column) and indeed achieve higher zero-shot scores than LISA-13B on PartImageNet and PascalParts. However, they still fall far short of PLUM in every metric, indicating that PLUM's special token-agnostic approach is more sample-efficient than the additional incorporationg of specialized tokens during segmenting LMM training.

## C.2 Effect of fine-tuning on Partonomy

Fine-tuning on only the Partonomy training split yields double-digit improvements across our baselines, with three consistent patterns emerging. PLUM (ft) attains the strongest macro-gIoU scores. It tops all three datasets in Identification and Intersection (e.g., 80.3 and 77.4 on PartImageNet), and remains within $\leq 2.0$ micro-gIoU of the best performer, demonstrating excellent part recall on rare classes. GLaMM (ft) is highly competitive on micro metrics. GLaMM (ft) slightly edges out PLUM on Difference-micro for PartImageNet and PascalParts. Nevertheless, the gap in macro-gIoU ($\geq 4.0$ gIoU) shows that GLaMM still under-segments uncommon parts. PixelLM-13B (ft) saturates early without Intersection gains.

## C.3 Dataset difficulty and domain shift

PACO_LVIS is the hardest split. Even after fine-tuning, macro-gIoU scores on PACO_LVIS are 20.0 to 30.0 points lower than on PartImageNet, reflecting its long-tailed distribution and heavy occlusions. PLUM's lead here (e.g., 9.0 macro-gIoU over GLaMM in Difference) underscores its robustness to open-vocabulary shift. PartImageNet rewards holistic part coverage. High Identification and Intersection numbers (e.g., 70.0 to 73.0 macro-gIoU for GLaMM/PLUM) reveal that most methods can capture coarse part extents when objects are canonical and well-cropped. Nonetheless, PLUM's advantage ($\approx 9.0$ macro-gIoU) suggests better treatment of fine-grained tails (e.g., bird beaks, insect legs). PascalParts exhibits the largest fine-tuning boost. All models gain > 38.0 macro-gIoU in Identification after fine-tuning—a potential consequence of its limited category set and high annotation quality. Here, PLUM (ft) again leads the macro-gIoU metrics, validating that its robustness.

Table 9: **Explanatory Part Segmentation's segmentation performance (gIoU) on PARTONOMY-PACO_LVIS.** "ft" = fine-tuned on Partonomy; *Part2Whole* and *Whole2Part* are not yet reported for this dataset.

| Methods | Extra Seg Data | Identification | | Intersection | | Difference | | Part2Whole | | Whole2Part | |
|---|---|---|---|---|---|---|---|---|---|---|---|
| | | micro | macro | micro | macro | micro | macro | micro | macro | micro | macro |
| LISA-13B | ✗ | 0.8 | 1.1 | 1.0 | 1.2 | 0.7 | 0.8 | 0.8 | 1.0 | 0.8 | 1.1 |
| PixelLM-13B | ✓ | 0.9 | 1.1 | 1.3 | 1.5 | 1.3 | 1.5 | 1.2 | 1.5 | 1.1 | 1.2 |
| GLaMM [37] | ✓ | 0.6 | 0.8 | 0.6 | 0.8 | 0.5 | 0.6 | 0.5 | 0.6 | 0.5 | 0.7 |
| **PLUM-13B** | ✗ | **6.7** | **16.0** | **8.2** | **17.3** | **5.2** | **14.3** | – | – | – | – |
| LISA-13B (ft) | ✗ | 29.0 | 30.7 | 30.6 | 32.4 | 26.8 | 28.0 | 16.9 | 15.8 | 28.4 | 30.0 |
| PixelLM-13B (ft) | ✓ | 37.8 | 40.5 | 34.8 | 42.7 | 36.0 | 36.5 | 6.8 | 8.5 | 6.8 | 8.4 |
| GLaMM (ft) [37] | ✓ | 36.8 | 39.3 | 38.9 | 41.2 | 34.7 | 36.7 | 17.4 | 16.0 | 31.0 | 33.3 |
| **PLUM-13B (ft)** | ✗ | 34.5 | **49.4** | 35.6 | **50.8** | 32.2 | **46.1** | – | – | – | – |

Table 10: **Explanatory Part Segmentation's segmentation performance (gIoU) on PARTONOMY-PartImageNet.** "ft" = fine-tuned on Partonomy; *Part2Whole* and *Whole2Part* are not yet reported for this dataset.

| Methods | Extra Seg Data | Identification | | Intersection | | Difference | | Part2Whole | | Whole2Part | |
|---|---|---|---|---|---|---|---|---|---|---|---|
| | | *micro* | *macro* | *micro* | *macro* | *micro* | *macro* | *micro* | *macro* | *micro* | *macro* |
| LISA-13B | ✗ | 7.4 | 8.7 | 7.4 | 8.5 | 5.0 | 5.7 | 7.3 | 8.5 | 7.1 | 8.3 |
| PixelLM-13B | ✓ | 8.0 | 9.4 | 10.3 | 12.8 | 1.5 | 1.4 | 1.7 | 1.8 | 8.4 | 10.1 |
| GLaMM [37] | ✓ | 5.0 | 6.1 | 5.3 | 5.9 | 3.9 | 4.6 | 4.8 | 5.7 | 4.4 | 5.3 |
| **PLUM-13B** | ✗ | **25.6** | 11.6 | **26.9** | 12.2 | **21.7** | 14.0 | – | – | – | – |
| LISA-13B (ft) | ✗ | 65.6 | 67.7 | 68.4 | 70.4 | 60.8 | 61.5 | 62.9 | 65.0 | 65.3 | 67.3 |
| PixelLM-13B (ft) | ✓ | 21.5 | 35.7 | 26.2 | 39.6 | 18.2 | 28.7 | 60.4 | 63.0 | 61.2 | 64.7 |
| GLaMM (ft) [37] | ✓ | 68.9 | 70.8 | 71.2 | 73.1 | 64.7 | 65.4 | 61.1 | 64.0 | 62.9 | 65.9 |
| **PLUM-13B (ft)** | ✗ | 67.9 | **80.3** | 70.2 | **77.4** | 63.3 | 60.7 | – | – | – | – |

Table 11: **Explanatory Part Segmentation's segmentation performance (gIoU) on PARTONOMY-PascalParts.** "ft" = fine-tuned on Partonomy; *Part2Whole* and *Whole2Part* are not yet reported for this dataset.

| Methods | Extra Seg Data | Identification | | Intersection | | Difference | | Part2Whole | | Whole2Part | |
|---|---|---|---|---|---|---|---|---|---|---|---|
| | | *micro* | *macro* | *micro* | *macro* | *micro* | *macro* | *micro* | *macro* | *micro* | *macro* |
| LISA-13B | ✗ | 2.2 | 2.4 | 2.4 | 2.6 | 1.7 | 1.8 | 2.1 | 2.3 | 2.1 | 2.4 |
| PixelLM-13B | ✓ | 3.6 | 4.0 | 3.7 | 4.1 | 3.7 | 4.1 | 3.5 | 3.8 | 3.0 | 3.2 |
| GLaMM [37] | ✓ | 2.0 | 2.1 | 2.4 | 2.4 | 1.2 | 1.4 | 1.7 | 1.8 | 1.6 | 1.8 |
| **PLUM-13B** | ✗ | **10.8** | **18.0** | 8.8 | 11.6 | **11.4** | 10.7 | – | – | – | – |
| LISA-13B (ft) | ✗ | 42.9 | 44.6 | 44.2 | 46.4 | 38.2 | 39.8 | 37.4 | 38.7 | 42.5 | 44.3 |
| PixelLM-13B (ft) | ✓ | 39.9 | 54.1 | 38.8 | 55.0 | 38.8 | 55.0 | 37.6 | 51.5 | 40.9 | 42.5 |
| GLaMM (ft) [37] | ✓ | 48.8 | 50.6 | 49.8 | 52.0 | 44.6 | 45.8 | 40.0 | 39.9 | 42.2 | 41.7 |
| **PLUM-13B (ft)** | ✗ | 48.4 | **64.0** | 49.3 | **65.5** | 44.4 | **54.0** | – | – | – | – |

## C.4   Additional Ablation Studies

In addition to the main experiments, we conducted several ablations and follow-up analyses. These studies validate PLUM's architectural choices, loss formulation, robustness, and dataset design.

**Bidirectional attention for BIO tagging.**   We confirmed that bidirectional attention is critical for accurate span extraction. Removing it severely degraded "I" tag accuracy on both Partonomy–PartImageNet and RefCOCO.

Table 12: **Bidirectional attention ablation.** Removing bidirectionality collapses span tagging performance.

| Model | B-Acc | I-Acc | O-Acc |
|---|---|---|---|
| PLUM (bi) | 98.59 | 87.32 | 99.98 |
| PLUM (no bi) | 100.00 | 15.86 | 99.78 |
| PLUM (bi, RefCOCO) | 99.98 | 99.87 | 100.00 |
| PLUM (no bi, RefCOCO) | 6.68 | 4.92 | 99.98 |

**Loss function comparison.**   To isolate the impact of the Focal–Tversky loss (FTL), we retrained PLUM with Dice loss under identical conditions.

Table 13: **DICE vs. Focal–Tversky.** PLUM's gains stem from its architecture choices; FTL provides marginal improvement over DICE.

| Loss | micro-gIoU | macro-gIoU | B-Acc | I-Acc | O-Acc |
|---|---|---|---|---|---|
| DICE | 66.47 | 79.86 | 92.99 | 86.48 | 99.99 |
| Focal–Tversky | 67.90 | 80.30 | 98.59 | 87.32 | 99.98 |

**Component ablations.**   We ablated PLUM's span extractor (SE) and feedback loop (F). Both contribute independently and jointly yield the best results.

**Feedback loop robustness.** While conditioning on past inputs may theoretically introduce error accumulation, manual inspection showed that even when early masks were incorrect, PLUM's feedback loop still produced correct later masks by relying on text embeddings. The loop also reduced duplicate or overlapping predictions. For example, the model without feedback repeated the same region for a lizard's foot and tail, while PLUM with feedback predicted them distinctly.

**Computational cost of feedback loop.** Timing experiments show that mask prediction is dominated by the LLM's forward pass: averaged over 1000 examples, the LLM forward pass took 2.47 seconds, decoding without a feedback loop took .0097 seconds, and decoding with a feedback loop took .0162 seconds, a .2% increase compared to the language model's time for its forward pass.

**Loss weight sensitivity.** We varied the loss weights to verify stability with respect to $\lambda_{\mathrm{KL}}$ and $\lambda_{\mathrm{seg}}$. Performance changed modestly, confirming robustness.

Table 14: **Loss weight sensitivity.** PLUM is stable; slightly higher $\lambda_{\mathrm{KL}}$ improves performance.

| $\lambda_{\mathrm{KL}}$ | $\lambda_{\mathrm{seg}}$ | $\lambda_{\mathrm{BCE}}$ | LR | gIoU | cIoU |
|---|---|---|---|---|---|
| 2.0 | 2.0 | 0.1 | 3e-4 | 0.385 | 0.331 |
| 4.0 | 2.0 | 0.1 | 3e-4 | 0.489 | 0.506 |
| 5.0 | 2.0 | 0.1 | 3e-4 | 0.517 | 0.494 |
| 6.0 | 2.0 | 0.1 | 3e-4 | 0.527 | 0.501 |
| 8.0 | 2.0 | 0.1 | 3e-4 | **0.573** | **0.546** |

Overall, these ablations confirm that PLUM's architectural innovations—bidirectional span extraction, mask feedback conditioning, and Focal–Tversky optimization—jointly account for its strong fine-grained segmentation and reasoning capabilities with minimal computational overhead.

### C.5 PARTONOMY-Core Samples

Fig. 7 provides an example of each question type for the *Explanatory Part Segmentation* task from **PARTONOMY**-Core.

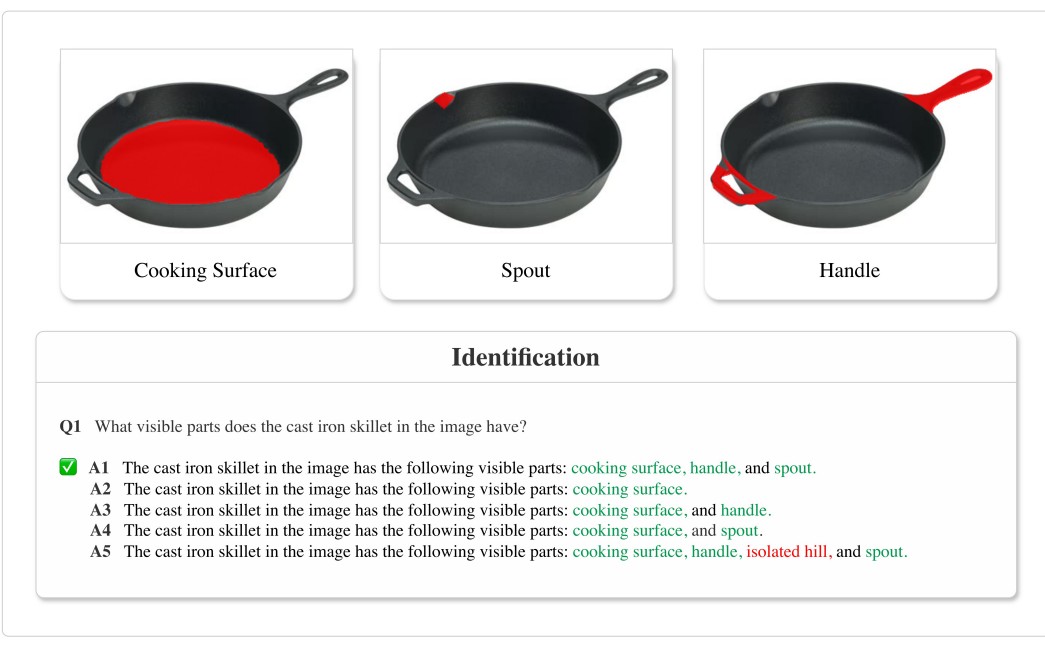

(a) An *Identification* question.

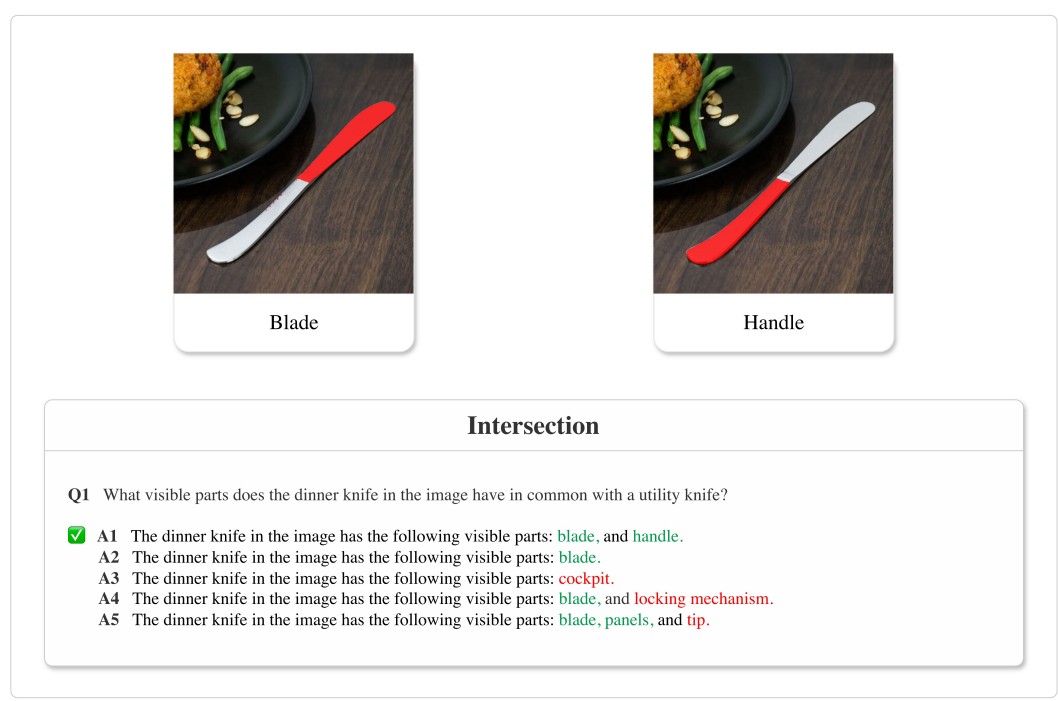

(b) An *Intersection* question.

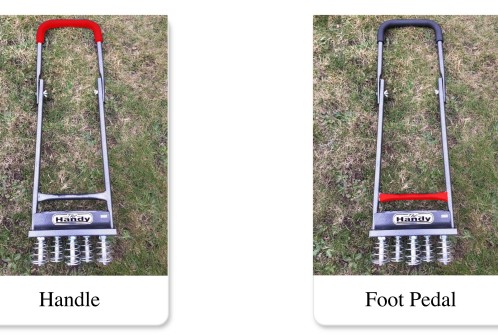
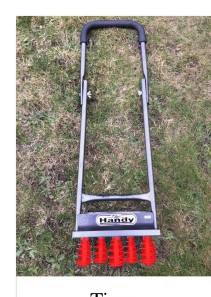

| Handle | Foot Pedal | Tines |
|--------|-----------|-------|

### Difference

**Q1** What visible parts does the aerating fork in the image have which a hole puncher does not?

✅ **A1** The aerating fork in the image has the following visible parts which a hole puncher does not: foot pedal, tines.
**A2** The aerating fork in the image has the following visible parts which a hole puncher does not: foot pedal, handle, and neck.
**A3** The aerating fork in the image has the following visible parts which a hole puncher does not: punch head, and tines.
**A4** The aerating fork in the image has the following visible parts which a hole puncher does not: foot pedal, and punch head.
**A5** The aerating fork in the image has the following visible parts which a hole puncher does not: tines.

(c) An *Difference* question.

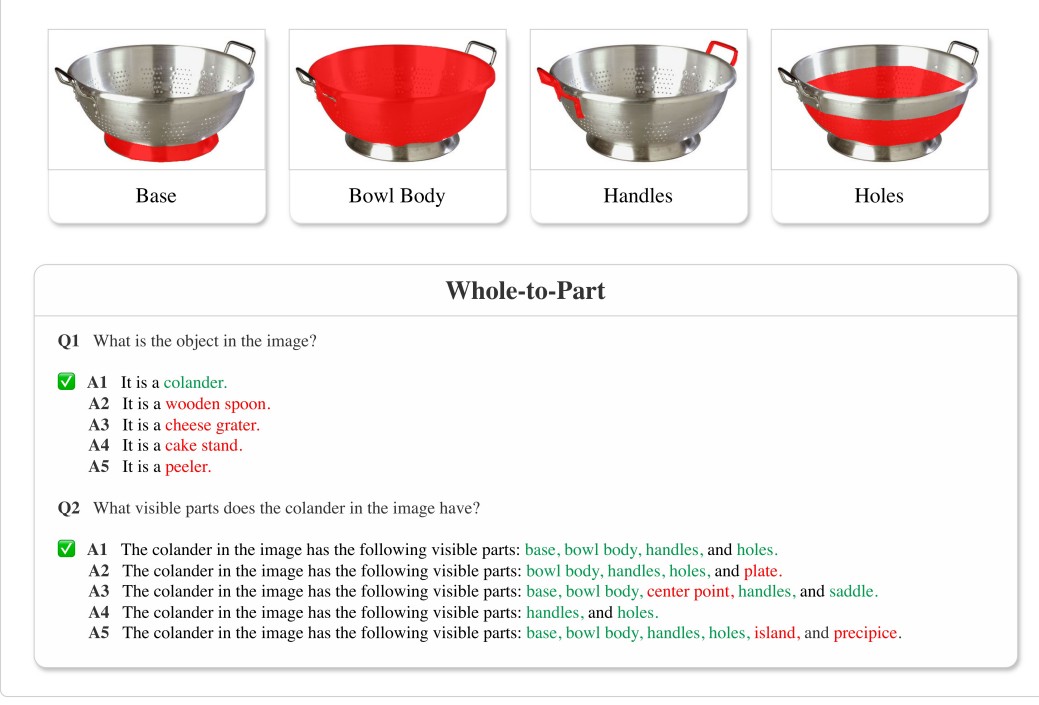

| Base | Bowl Body | Handles | Holes |
|------|-----------|---------|-------|

### Whole-to-Part

**Q1** What is the object in the image?

✅ **A1** It is a colander.
**A2** It is a wooden spoon.
**A3** It is a cheese grater.
**A4** It is a cake stand.
**A5** It is a peeler.

**Q2** What visible parts does the colander in the image have?

✅ **A1** The colander in the image has the following visible parts: base, bowl body, handles, and holes.
**A2** The colander in the image has the following visible parts: bowl body, handles, holes, and plate.
**A3** The colander in the image has the following visible parts: base, bowl body, center point, handles, and saddle.
**A4** The colander in the image has the following visible parts: handles, and holes.
**A5** The colander in the image has the following visible parts: base, bowl body, handles, holes, island, and precipice.

(d) A *Whole-to-Part* question.


Figure 7: Examples of the different question types from the Partonomy-Core dataset.

