# OpenReview forum: "PARTONOMY: Large Multimodal Models with Part-Level Visual Understanding"
_NeurIPS.cc/2025/Conference — NeurIPS 2025 spotlight_

### Official Review · Reviewer_Ke3p · 2025-06-28

**Clarity:** 3
**Significance:** 4
**Originality:** 3
**Rating:** 5
**Confidence:** 4

**Summary:**

This paper addresses a key limitation of large multimodal models (LMMs): their inability to perform fine-grained part-level visual understanding. They propose a dataset/benchmark PARTONOMY  for "Explanatory Part Segmentation," evaluating LMMs on identifying object parts, comparing parts across objects, and reasoning about part-whole relationships. It spans 787 objects and 2,768 part labels, with diverse coverage beyond prior datasets. A part-level segmenting LMM PLUM is proposed to improve part-level understanding by avoiding special [SEG] tokens (via span tagging) and using a feedback loop to refine segmentations with prior masks. PLUM outperforms state-of-the-art LMMs on part segmentation tasks while retaining strong performance on general vision-language tasks.

**Questions:**

1. PLUM’s feedback loop uses FiLM layers to condition on prior masks. How are multiple prior masks (e.g., for overlapping parts) weighted or disambiguated? Does the loop ever propagate errors (e.g., a poorly segmented first part worsening subsequent masks)?
2. The authors use Llama 2-70B to generate part labels for PARTONOMY-Core, then manually prune "non-visible" parts. How were disagreements between annotators resolved? Was inter-annotator agreement measured?

**Ethical Concerns:**

["NO or VERY MINOR ethics concerns only"]

**Final Justification:**

The authors have addressed my concerns in detail and I would like keep my original rating as 5.

**Limitations:**

Yes; the authors partially addressed limitations in the paper. For additional limitations, please refer to the "Weaknesses" section.

**Quality:**

4

**Strengths And Weaknesses:**

Strengths:
1. The benchmark and model are rigorously designed. PARTONOMY-Core’s manual annotation ensures high-quality part labels, and PLUM’s architecture (span tagging, feedback loop) is theoretically motivated to mitigate distribution shift and improve segmentation consistency.
2. Part-level understanding is a foundational yet underexplored challenge for LMMs. PARTONOMY fills a critical need for standardized evaluation, while PLUM provides a novel architectural paradigm to improve grounding without sacrificing general reasoning.
3. The span tagging mechanism (replacing [SEG] tokens) and mask feedback loop are innovative solution.
4. The paper is well-written and easy to follow.


Weaknesses:
1. The FiLM-based feedback loop’s implementation details are vague. For example, how prior masks are encoded into feature maps and pooled via "patch-wise attention" is not fully explained.
2. PLUM’s feedback loop and span tagging may increase computational cost, limiting real-world deployment.

---

> ### Author Rebuttal · Authors · 2025-07-31
>
> We sincerely appreciate the time and effort you have dedicated to reviewing our work. In the following, we have provided our response to your questions and comments for clarification.
>
> > **W1: The FiLM-based feedback loop’s implementation details are vague.**
>
> We elaborate the two specific components in our proposed Mask Feedback Loop mechanism down below:
> (i) **FiLM**, in the original paper, was introduced to condition the visual features in the intermediate layers on latent text representations, which is specifically what we used the FiLM layer for in PLUM to condition the mask region representation on the text span embeddings. This enables us to retain the text-conditioned mask representations throughout the feedback loop.
> (ii) **Patch-wise Attention Pooling** was used in PLUM to temporally incorporate information across all prior masks; meaning, we use the attention pooler to encode the previously generated masks before we pass them into the mask prompt encoder.
>
> Specifically: we modify SAM's prompt encoder by injecting FiLM layers between its convolutional layers. We encode each mask using the modified prompt encoder and condition its FiLM layers using the text embeddings corresponding to each mask. This yields a set of feature maps, one for each mask. As SAM's mask decoder takes a single feature map as its mask prompt, we then perform a patch-wise attention pool to aggregate information across all feature maps into a set of learnable CLS embeddings. These CLS embeddings constitute the feature map that is passed into SAM's mask decoder as the mask prompt.
>
> ---
>
> > **W2: PLUM’s feedback loop and span tagging may increase computational cost, limiting real-world deployment.**
>
> Span tagging happens in parallel (a single forward pass over the embeddings). Our feedback loop is a straightforward approach that improves performance—however, it is true that parallelizing our current feedback loop design isn't possible, suggesting an architecture which decodes in parallel, performs joint attention over predictions, and refines the masks based on the attentional outputs in parallel would be a promising direction for future research. However, timing experiments show that mask prediction is dominated by the LLM forward pass: averaged over 1000 examples, the LLM forward pass took 2.47 seconds, decoding without a feedback loop took .0097 seconds, and decoding with a feedback loop took .0162 seconds, a .2% increase compared to the language model's forward pass time.
>
> > **Q1: How are prior masks weighted or disambiguated? Error Propagation?**
>
> Prior masks are not manually weighted in the patch-wise attention pool—instead, the learned CLS tokens (each corresponding to a patch) each perform attention across the patches for the embedded masks. We do not disambiguate between mask embeddings (e.g. we do not add temporal embeddings) as we believe that temporal decoding order should not affect performance.
>
> Autoregressive models like our feedback loop can indeed introduce error accumulation. However, we manually inspected samples with and without the feedback loop and did not observe significant error accumulation. In fact, both models seemed to make very similar mistakes, sometimes incorrectly predicting the first mask but going on to predict the remaining ones correctly. Differences arose when the feedback loop predicted a larger object area, as future masks were less likely to be contained in that previously predicted object area.
>
>
> > **Q2: How was Annotator Agreement Resolved?**
>
> As noted, our ontology construction process started with Llama-2-70B proposing a set of parts (especially useful for highly technical objects, like military vehicles), then an initial pruning occurs based on individual research. However, annotators do not know with certainty whether a part is commonly present in images of the object until they start annotating them. Therefore, we do not measure inter-annotator agreement for part pruning, but instead did a three-round filtering process including the initial pruning after research, then large-scale part annotation with further pruning, and finally a third filtering/refinement pass where annotations are refined and parts not found in the images are dropped.

---

> > ### Comment · Reviewer_Ke3p · 2025-08-05
> >
> > Thank you for your detailed response, which has addressed my concerns. Therefore, I would like to keep my rating as 5.

---

> > > ### Author Response · Authors · 2025-08-05
> > >
> > > Thank you again for your thoughtful feedback, and for taking the time to read our response.

---

### Official Review · Reviewer_bML3 · 2025-07-01

**Clarity:** 4
**Significance:** 3
**Originality:** 3
**Rating:** 5
**Confidence:** 5

**Summary:**

This paper introduces a new benchmark called PARTONOMY, which requires Large Multimodal Models (LMMs) not only to identify the generic parts of an object but also to learn the part-whole relationships between entities. Experiments on baseline models such as LISA show that current methods fail on this benchmark, thereby highlighting the gap in their part-grounding abilities. Furthermore, the authors address two weaknesses of existing LMMs in segmentation tasks: the need for an additional special token and the auto-regressive paradigm. They propose a new LMM called PLUM that uses span tagging rather than a special token and conditions its predictions on prior ones in a feedback loop. Several experiments on different tasks show PLUM's outstanding performance.

**Questions:**

PLUM generates part masks serially, one by one. It is questionable whether this is the most efficient approach. For a prompt like, "Please segment the airplane's wings, engines, and tail," could the model first fully comprehend all target parts from the text and then generate all masks in parallel? Parallel decoding could potentially be faster and might avoid the error accumulation issue mentioned above. The authors should provide a stronger justification for adopting a serial generation approach.

**Ethical Concerns:**

["NO or VERY MINOR ethics concerns only"]

**Final Justification:**

The author has addressed my concerns about the insufficient benchmark and my confusion regarding the design of the Mask Feedback Loop. I think the paper will be ready for acceptance once the author incorporates the discussed revisions into the final version. Therefore, I have decided to raise my score.

**Limitations:**

yes

**Quality:**

4

**Strengths And Weaknesses:**

## Stengths

1. The proposed PARTONOMY benchmark provides a valuable research topic and resource for the community. It effectively demonstrates a key weakness of existing LMMs: their failure to identify part-whole relationships.

2. The observation of the two weaknesses is insightful and correct. The proposed methods address these issues to a certain extent.

3. Using span tagging instead of a `[SEG]` token avoids the need to train an additional special token, which can be beneficial for fine-tuning with limited data.

4. Introducing past information as a condition is a good idea to enhance the segmentation ability of LMMs.

5. Experiments show that PLUM achieves better performance on various tasks, even when using less data compared to its competitors.

## Weaknesses

1. A dataset of 1k images may seem insufficient for a benchmark. I am wondering if the authors can prove that 1k images are enough to robustly evaluate LMMs. Specifically, the authors could evaluate models using varying numbers of images from PARTONOMY and plot a performance curve to determine the point at which the results converge, thereby demonstrating the sufficiency of the benchmark size.

2. The second problem addressed by the authors is very similar to that in paper [1], which also introduces entity relationships into LMMs. It would be better to discuss the relationship with this paper and include it as a baseline for comparison.

3. PLUM designs a Mask Feedback Loop to use historical masks as conditions to enhance future predictions; however, this inevitably introduces error accumulation. Specifically, if the model outputs a wrong or inaccurate mask, subsequent predictions will be dynamically affected. The paper should discuss this phenomenon.

4. The authors frame this as an entirely new task. However, it could essentially be considered a more challenging variant of "Referring Expression Segmentation" that involves complex reasoning, or a form of "Multi-object Segmentation." The core objective remains the segmentation of specific image regions based on textual instructions. While the question formats (e.g., Part Comparison, Part-Whole Reasoning) are well-designed, defining this as a completely new and independent task category may be an overstatement.

[1] SegLLM: Multi-round Reasoning Segmentation with Large Language Models

---

> ### Author Rebuttal · Authors · 2025-07-31
>
> We sincerely appreciate the time and effort you have dedicated to reviewing our work. In the following tables, we have provided our response to your questions and comments.
>
> > **W1: A dataset of 1k images may seem insufficient for a benchmark.**
>
> We appreciate the concern about the small evaluation set. We would like to note that in addition to Partonomy-Core, we also provide evaluation results on holdout sets of the other part datasets (Figure 4). Further, 1,000 examples correspond to ≈ 5K segmentation masks. Additionally, past benchmarks have far fewer examples (e.g., LISA's ReasonSeg benchmark which defines the task has only 200 evaluation samples). Our goal was to provide a wide range of highly technical parts and concepts, instead of providing more examples of fewer parts (an easier annotation process) as in previous benchmarks.
>
> Following the suggestion to compute performance on varying amounts of evaluation data, we evaluate the finetuned LISA model on varying percentages of our 1K Partonomy-Core examples. We see that results are relatively stable across evaluation sizes.
>
> | Model (% Eval Dataset) | Identification | Intersection | Difference |
> | - | - | - | - |
> | LISA ft (20%) | 34.7 / 35.9 | 38.3 / 40.0 | 30.4 / 32.8 |
> | LISA ft (40%) | 32.7 / 34.2 | 37.0 / 37.8 | 29.0 / 29.4 |
> | LISA ft (60%) | 33.9 / 34.1 | 38.0 / 39.5 | 31.3 / 32.7 |
> | LISA ft (80%) | 33.2 / 35.2 | 37.0 / 38.2 | 30.7 / 31.8 |
> | LISA ft (100%) | 33.6 / 35.4 | 37.0 / 38.4| 30.4 / 31.6 |
>
> ---
>
> > **W2: Similar Motivation to SegLLM**
>
> Thank you for referring us to the work. SegLLM indeed has a similar architectural motivation in noting that existing segmenting LMMs are unable to refer to past mask predictions. However, the primary motivation of SegLLM is to perform multi-round conversations where the user refers to previous objects—SegLLM is even trained with an explicit loss function for the referent mask. Our goal, in contrast, is to broadly enable mask decoding within the context of prior masks, applying this to part segmentation. This is reflected in our architectural change without any additional objective function.
>
> Some core differences are as follows:
> - SegLLM passes previously encoded masks through the entire LLM. In contrast, we encode prior masks (conditioned on their corresponding text) and pass them right back into the mask decoder without having to propagate through the LLM backbone. While their approach is more expressive (having passed the masks through the LLM), it is much more computationally expensive: SegLLM must perform as many LLM forward passes as there are [SEG] tokens, whereas ours only requires a single LLM forward pass.
> - Each of SegLLM's mask embeddings require a forward pass of the masked image through a CLIP vision encoder and a projector. Ours, by contrast, takes the low-resolution predicted masks and encodes them with SAM's lightweight prompt encoder.
> - SegLLM is trained with teacher forcing on the segmented masks. Ours is trained directly on its generated predictions which may make it more robust to poor mask predictions.
> - SegLLM is trained with a loss function that explicitly forces the model to decode referenced masks. Our decoding process requires no
>
> We will include a discussion of SegLLM in the related works and would be happy to train it as an additional baseline due to its similarity.
>
> ---
>
> > **W3: PLUM designs a Mask Feedback Loop Introducing Error Accumulation:**
>
> It is true that the common characteristic of conditioning on the past predictions to generate the current output (e.g., autoregressive text generation) can introduce error accumulation. However, we would like to underscore the great success with models which decode autoregressively (e.g. LLMs) and while a possibility, it is not a guarantee that earlier errors will affect future ones (unless the earlier predictions are significantly off-the-mark, which we have not observed yet).
>
> While theoretically plausible, we manually inspected samples with and without the feedback loop and did not observe significant error accumulation. In fact, both models seemed to make very similar mistakes, sometimes incorrectly predicting the first mask but going on to predict the remaining ones correctly. Differences arose when the feedback loop predicted a larger object area, as future masks were less likely to be contained in that previously predicted object area.
>
> We will discuss these analyses and drawbacks in our limitations and supplementary material.
>
> ---
>
> > **W4: Defining Explanatory Part Segmentation as an entirely new task is an overstatement.**
>
> We do agree that our proposed Explanatory Part Segmentation task can be viewed as a variant of previous segmentation tasks in some way. We, nevertheless, would like to kindly note that our proposed task is beyond simple grounding of textual phrases, but requires implicit visual reasoning of part-whole relations across different visual objects. Furthermore, our task is notably distinct in its level of challenge (e.g. even though the pretrained LMMs are trained on part datasets, their performance remains low), and our question formulation requiring capabilities beyond pixel-level grounding (e.g. requiring reasoning over intersection, difference, and part-to-whole) make it unique from the existing referring segmentation landscape. We will note our task's relationship to referring segmentation in the related works.
>
> ---
>
> > **Q: Generating part masks serially may not be the most efficient**
>
> Indeed, generating the part masks serially is slower than parallel decoding. However, our experiments show that the feedback loop improves performance, and we would argue that the benefits outweight the costs. Further, timing experiments show that mask prediction is dominated by the LLM forward pass: averaged over 1000 examples, the LLM forward pass took 2.47 seconds, decoding without a feedback loop took .0097 seconds, and decoding with a feedback loop took .0162 seconds, a .2% increase compared to the language model's forward pass time.
>
> An architecture that performs joint attention over predictions and refines the masks based on the attentional outputs in parallel would be a promising direction for future research.

---

> > ### Comment · Reviewer_bML3 · 2025-08-06
> >
> > Thanks for your reply, it cleared up most of my concerns.
> >
> > I'm still thinking about the error accumulation problem, though. The paper says the model uses the last result to improve the next one, which makes sense. But it also makes me wonder: what happens if an early prediction is just wrong?
> >
> > You mentioned in the rebuttal that the model could mess up the first mask but still predict the later ones correctly. To be honest, that feels a bit counter-intuitive. It seems like that would only work if the predictions don't depend on each other that much.
> >
> > I think it would really help if you could add a simple experiment in the revision to show how this works. For example, you could try manually feeding it an incorrect mask to start with and show us what happens. I think that would make your claim much stronger.

---

> > > ### Author Response · Authors · 2025-08-06
> > > **Response to Reviewer bML3's Comment**
> > >
> > > Thank you very much for your insightful questions! We agree that the idea that the feedback loop provides past predictions to inform future ones, and the observation that the model can make early mistakes while still correctly predicting later parts, appears contradictory. The latter observation suggested to us that the decoder was relying more on the text embeddings to make its predictions than on past masks. The scenarios where different early predictions would result in different subsequent ones were rarer. This does suggest that not all parts are highly correlated. For example, an airplane's wings, engines, and body have minimal overlap and can be independently identified, even if, say, the wings are misidentified at the start (and therefore would not provide a correct signal for where the engines should be).
> > >
> > > Your suggested experiment would shed light on how robust the feedback loop is to a poor initial segmentation—for example, by providing a randomly generated mask, blank mask, or the ground truth of the second mask to be predicted. The latter experiment in particular may indicate whether the model is less likely to predict previously predicted regions (though this may still be affected by each mask's modulating text). We will gladly run these experiments to better understand the feedback loop's behavior and add them to our revision.

---

> > > > ### Comment · Reviewer_bML3 · 2025-08-06
> > > >
> > > > Thank you for the clarification.
> > > >
> > > > I am trying to reconcile two points. Your paper states on line 50-51: "Second, these segmenting LMMs discard their predictions after each output, missing the opportunity to incorporate prior information during the decoding process." and claim that conditioning on past predictions is a key benefit of your proposed method.
> > > >
> > > > However, in your rebuttal, you mentioned: "The latter observation suggested to us that the decoder was relying more on the text embeddings to make its predictions than on past masks."
> > > >
> > > > These two statements seem contradictory. Could you please provide further evidence to demonstrate the tangible benefits of conditioning on past predictions in your model?

---

> ### Author Response · Authors · 2025-08-07
> **Follow-up Response to Reviewer bML3's Comment**
>
> Thank you for your questions. We understand why these statements might seem contradictory at first and would like to provide further clarification on the feedback loop.
>
> First, we noted based on our manual inspection of model predictions that the majority of the prediction signal appeared to come from the text embeddings. We drew this conclusion as the model was able to recover from poor initial predictions, going on to segment later parts correctly. This scenario is more common when the initial incorrect segmentation is loosely correlated with later parts. For example, the "openable nose" of an airplane does not give a strong signal for where its "engines" are. Even if the model predicted the mask for "openable nose" wrong, the model was able to correctly predict subsequent masks for distinct parts, e.g., "engines". The feedback loop's ability to recover suggests that the text embeddings constitute a significant portion of the segmentation signal.
>
> However, the feedback loop does provide models with the *opportunity* to incorporate past predictions if they would be helpful. We see qualitative evidence that the feedback loop conditions future predictions on past ones. We note two key benefits:
>
> (i) *Better localization of overlapping parts*: If an early prediction covers most of an object, later ones tend to be more accurate, especially in cases where the parts are not easily localizable. One example was of an image of a turtle on a forest floor. Both models (with and without the feedback loop) correctly predicted the turtle's body, but only the one with the feedback loop was able to identify its barely-visible head. The one without the feedback loop selected a leaf on the ground, unable to use the prior prediction of the turtle's body to constrain its search.
>
> (ii) *Reduction of duplicate masks*: The feedback loop was also less prone to output the same prediction for multiple parts. One notable example was on an image of a lizard. The model without the feedback loop correctly segmented the lizard's feet then incorrectly predicted one of its feet again as the lizard's tail, unable to see that it had already identified that region. The model with the feedback loop, having also correctly segmented the lizard's feet, did not repeat its prediction, instead correctly segmenting the lizard's tail.
>
> We also underscore quantitative evidence below (as an extension of Figure 5a) that suggests that the feedback loop provides a tangible benefit (*SE* refers to Span Extractor and *F* refers to Feedback Loop):
> |Model | micro-gIoU | macro-gIoU|
> |-- | -- | -- |
> |LISA *(-SE, -F)* | 65.6 | 67.7 |
> |LISA + Feedback Loop *(-SE, +F)* | 66.7 | 68.6 |
> |PLUM - Feedback Loop *(+SE, -F)* | 61.4 | 73.9 |
> |PLUM *(+SE, +F)* | 67.9 | 80.3 |

---

> > ### Comment · Reviewer_bML3 · 2025-08-07
> >
> > Thank you for the clarification and additional experiment, which have resolved my main concerns; I will adjust my score accordingly and ask that you please include the relevant discussion in the final version of the paper.

---

> > > ### Author Response · Authors · 2025-08-07
> > > **Appreciation for Active Engagement**
> > >
> > > Dear reviewer, thank you for your active engagement and thoughtful comments on our paper. We will include our discussion in the final revision of our paper.

---

### Official Review · Reviewer_KqXb · 2025-07-03

**Clarity:** 4
**Significance:** 3
**Originality:** 3
**Rating:** 5
**Confidence:** 4

**Summary:**

The paper investigates fine-grained part understanding with large multimodal models (LMMs). In doing so, the authors first introduce a pixel-level part grounding benchmark, PARTONOMY, which contains a training split and a test split, aiming to examine the capabilities of LMMs for part grounding, part comparison, and part-whole relationship reasoning. Moreover, they identify two issues with existing part grounding models: a distribution shift due to the reliance on a special segmentation token and a lack of modeling history part groundings. They further address the issues using an improved pre-training strategy and deliver a more performant LMM: PLUM. The experimental results demonstrate the superior performance of PLUM.

**Questions:**

Text evaluation in Table 3 shows marginal performance gaps between different models. Does that mean the answer choices construction is suboptimal? The authors argue that "each answer choice shares extensive lexical overlap." Is it possible to quantify this, and are there any plans to resolve this issue?

**Ethical Concerns:**

["NO or VERY MINOR ethics concerns only"]

**Final Justification:**

accept. a solid work and rebuttals cleared my concerns.

**Limitations:**

1. \# images and segmentation masks in Table 1 seem missing.

**Paper Formatting Concerns:**

not that i am aware of

**Quality:**

3

**Strengths And Weaknesses:**

## strengths

1. The paper addresses an important problem: pixel-level part understanding, which has been largely overlooked in LMM research.
2. The authors introduce a benchmark and a set of evaluation metrics for part understanding.
3. A new training paradigm is proposed to address the limitations of existing segmentation LMMs in part grounding.
4. A new LMM, PLUM, which is trained using the proposed learning paradigm, is delivered and further demonstrates strong part-understanding performance.

## weeknesses

1. The chosen baseline open-vocabulary segmentation models seem outdated. Here are some more recent models: Grounded SAM/DINO: https://github.com/IDEA-Research/Grounded-SAM-2; https://arxiv.org/abs/2412.16334v1
2. Lacking comparison with frontier models like GPT-4o. A potentially stronger baseline could be a combination of pre-trained VLMs for visual reasoning and Grounded SAM/DINO for part grounding. Including this baseline can better justify the superiority of the proposed unified model.
3. Lacking an in-depth analysis of the span extractor. The incorporation of a specific span extraction module might complicate the model design. VLM should be able to identify segmentation-related text spans or can be trained to predict them directly.

---

> ### Author Rebuttal · Authors · 2025-07-31
>
> We sincerely appreciate the time and effort you have dedicated to reviewing our work. In the following, we have provided our response to your questions and comments for clarification.
>
> > **W1: Outdated Open-Vocabulary Segmentation Model:**
>
> While our work focuses on segmenting LMMs and the performance of open-vocabulary segmentation models does not change our conclusions, we agree that the comparison provides helpful context. We actually played with Grounded-SAM in preliminary experiments but chose to investigate other models due to Grounded-SAM's poor performance. The DINO.txt model was not released at the time of our paper's submission, but we agree it appears promising. We value the suggestion to include newer baselines and will provide additional results on newer open-vocabulary segmentation models in the camera-ready.
>
> ---
>
> > **W2: Lacking comparison with frontier models like GPT-4o:**
>
> In Table 2, the part grounding using open-vocabulary segmentation model is done by providing these models with the ground-truth (gt) part text labels. Meaning, their results in Table 2 can be seen as an upper bound for GPT-4o pipelined with X-Decoder, SEEM, or GroundedSAM/DINO (and other additional open-vocabulary segmentation models). Nonetheless, we do agree that this experiment could better highlight the need for a unified, end-to-end model such as PLUM, so we will include additional experiments on GPT-4o augmented with open-vocabulary segmentation models in our upcoming camera-ready manuscript.
>
> ---
>
> > **W3: Lacking an in-depth analysis of the span extractor**
>
> We agree that performing tagging on top of the LMM requires additional data preprocessing to ground the answer in the original text. However, we would like to respectfully point out that reducing the LMM's distribution shift and the subsequent performance gains pretrained PLUM obtains over its peers justifies the addition of the span extraction module. To further analyze our span extractor, we provide additional experiments in the following tables:
>
> To demonstrate the importance of bidirectional attention, we train PLUM with and without the bidirectional self-attention layer for BIO tagging. We finetune on the training split of Partonomy-PartImageNet and evaluate on its validation set.
>
> |                   | b-acc | i-acc | o-acc |
> |-------------------|-------|-------|-------|
> | PLUM (w/ bidirectional layer) | 98.59 | 87.32 | 99.98 |
> | PLUM (w/o bidirectional layer) | 100.00 | 15.86 | 99.78 |
>
> The results show that the bidirectional layer plays an essential role in BIO tagging, and thus the span extraction. The result suggests that the bidirecitonal layer plays a critical role in identifying the text spans (as shown by the `i-acc`), while `b-acc` appears to be high in both settings. We hypothesize that finetuning on a fixed pattern of answers in Partonomy, e.g., `ASSISTANT: frame, handlebars, lock, and seats` across the dataset should have given the easy positional bias for the model to exploit, achieving good accuracy for `B` class. Nonetheless, the substantial loss of accuracy for the `I` class shows that the bidirectional layer is an essential part of our proposed PLUM model's span extraction capability.
>
> We also provide additional results for Referring Expression Segmentation to demonstrate the importance of employing the bidirectional attention layer as the span extractor to identify the textual span for segmentation. We note that referring expression segmentation is more challenging than simple part text tagging as it requires highlighting longer text spans for segmentation (Evaluation was performed on RefCOCO | Unc | Val set; “ft” indicates finetuned on RefCOCO training dataset):
>
> | | b-acc | i-acc | o-acc |
> |-|-|-|-|
> | PLUM (w/ bidirectional layer) (ft) | 99.98 | 99.87 | 100.00 |
> | PLUM (w/o bidirectional layer) (ft) | 6.68  | 4.92  | 99.98  |
>
>
> ---
>
> > **Question: Text evaluation in Table 3 shows marginal performance gaps between different models.**
>
> Thank you for your question. We would like to kindly note that the high lexical overlap among answer choices is intentional. Our goal was to keep each answer choice as similar as possible to force the model to pay attention to small changes in the parts (e.g., “wings, landing gear and propulsion component” vs. “wings, guidance system and propulsion component”) when determining the most probable answer choice. The low accuracy and high variance in Table 3 suggest that the next-token prediction objective is insufficient for part detection and to establish part-whole relationships between objects and their constituent parts. This indicates a promising area for future research (e.g. contrastive losses on close answer choices).
>
> To quantify the overlap, we tokenized each answer choice and computed the average IoU between the correct answer choice and the most similar wrong answer choice. If we consider all tokens in each answer, we obtain an overlap of 89.43% (on average, the most similar answer choice has 89% of the same tokens as the correct answer). If we only consider the parts, we obtain an overlap of 41.01% (on average, the most similar answer choice has 41% of the same parts as the correct answer). An example of a challenging set of answer choices is below. Here, the first answer choice is the correct answer.
>
> ```
> > "The dirt bicycle in the image has the following visible parts: frame, handlebars, knobby tires/wheels, pedals, and seats.",
> > "The dirt bicycle in the image has the following visible parts: frame, handlebars, lock, and seats.",
> > "The dirt bicycle in the image has the following visible parts: frame, grip, handlebars, knobby tires/wheels, pedals, seats, and tires/wheels.",
> > "The dirt bicycle in the image has the following visible parts: antennas, frame, handlebars, knobby tires/wheels, pedals, and seats.",
> > "The dirt bicycle in the image has the following visible parts: frame, handlebars, knobby tires/wheels, pedals, pivot, pop filter, and seats."
> ```
> ---
>
> > **Limitations: # images and segmentation masks in Table 1 seem missing.**
>
> Thank you for pointing out the omission. The numbers are as follows: the number of images = 74,500, and the number of segmentation masks = 407,101. We have updated Table 1 with these values.

---

> ### Comment · Reviewer_KqXb · 2025-08-06
>
> thank you for your responses. i look forward to the updates you promised to make in the final version.

---

> > ### Author Response · Authors · 2025-08-06
> > **Response to Reviewer KqXb's Comment**
> >
> > Thank you for your active engagement with our paper and for sharing a thoughtful, considerate review!

---

### Official Review · Reviewer_gK9E · 2025-07-11

**Clarity:** 2
**Significance:** 3
**Originality:** 3
**Rating:** 4
**Confidence:** 3

**Summary:**

This paper introduces Partonomy, a new pixel-level part grounding benchmark designed to evaluate the ability of models to reason about part-whole relationships. The dataset covers a broad range of object categories (787 in total), including common items, biological entities, and specialized objects, and is annotated with 1,157 fine-grained part labels. A key feature of this benchmark is its requirement for models to not only recognize part-whole relationships via textual description, but also justify them through precise visual segmentations.

Building upon this dataset, the authors propose PLUM, a segmentation-enabled large multimodal model (segmenting LMM) that employs span tagging to avoid distribution shift caused by [SEG] token. Moreover, it refines the segmentation prediction via a feedback loop.

Experimental results demonstrate that existing state-of-the-art LMMs exhibit limited capabilities in fine-grained part grounding. In contrast, the proposed PLUM model shows compelling performance on the Partonomy benchmark as well as other evaluation tasks involving reasoning, segmentation, VQA, and visual hallucination.

**Questions:**

Please review the weaknesses outlined above. In particular, I would like to highlight the following four key questions:
- Could you provide theoretical justification or empirical evidence to support the claim that “compared to causal attention, bidirectional attention is more critical for reliable BIO span tagging”?
- Could you conduct a fair comparison using Dice loss within PLUM to isolate the impact of the proposed Focal-Tversky Loss?
- Could you provide a comprehensive ablation study of the three core components in PLUM: Span Extractor, KL Constraint, and Mask Feedback Loop?
- Could you include an evaluation of the Span Extractor’s classification performance for the {B, I, O} labels?

**Ethical Concerns:**

["NO or VERY MINOR ethics concerns only"]

**Final Justification:**

The authors have addressed my primary concerns. Although I remain unconvinced by their explanation of **W4** (motivation and fairness of using the Focal-Tversky loss), this is not a major issue. Overall, I am inclined to accept the paper and have increased my rating accordingly.

**Limitations:**

No, the authors did not discuss the limitations and potential negative societal impact of their work.

**Paper Formatting Concerns:**

No, there is no major formatting issues in this paper.

**Quality:**

2

**Strengths And Weaknesses:**

**Strengths:**

The proposed explanatory part segmentation task and the corresponding Partonomy dataset are valuable contributions. Also, the taxonomy introduced for the explanatory part segmentation task is comprehensive and well-structured.

**Weaknesses:**

My primary concerns span the areas of data, method, experiments, and clarity. Specific concerns are as follows:

**Data:**

- **W1. Lack of Demo Samples from Partonomy / Partonomy-Core.** Given that the explanatory part segmentation task and the Partonomy dataset are the core contributions of the paper, it is essential to provide representative samples in the supplementary materials to help readers better understand their details and annotation quality.

**Method:**

- **W2. Unverified Claim Regarding Bidirectional Attention (L220–221).** The paper claims that “bidirectional attention is critical for reliable BIO span tagging” but provides no theoretical justification or empirical evidence to support this claim.
- **W3. Unclear Roles of the Components in Mask Feedback Loop.** The proposed mask feedback loop involves additional components to the SAM decoder, including Feature-wise Linear Modulation (FiLM) and Patch-wise Attention Pool. However, it is still unclear why these two specific techniques are involved and how they influence the overall performance. Can FiLM be replaced by alternatives, such as a standard cross-attention mechanism?
- **W4. Use of Focal-Tversky Loss Lacks Justification.** As all baseline methods are evaluated using Dice loss while PLUM uses a different loss, it is unclear whether observed performance gains stem from architectural innovations or the loss function itself. A comparative study using Dice loss within PLUM is necessary to isolate the contribution of each component.

**Ablation Study:**

- **W5. Incomplete Ablation of Core Components.** The effectiveness of PLUM is attributed to three key components: Span Extractor, KL Constraint, and Mask Feedback Loop. However, a comprehensive ablation across all 8 combinations of these components is missing. The ablation study should be conducted not only on Partonomy-Core but also across the other evaluation tasks (e.g., reasoning segmentation, VQA, and hallucination detection).
- **W6. Lack of Analysis on Loss Weights.** The paper explores the KL constraint weight ($\lambda_2$) but does not examine the influence of other loss terms, particularly span extraction loss ($L_{span}$) and segmentation loss ($L_{segm}$). A more thorough loss weight sensitivity analysis is needed.

**Other Experiments:**

- **W7. No Evaluation of Span Extractor’s {B, I, O} Classification.** The quality of the segmentation largely depends on accurate BIO tagging from the Span Extractor. However, no explicit evaluation (e.g., accuracy, F1 score) is provided for this three-way classification task.
- **W8. Misleading Conclusion in L268–273.** The authors claim that “Table 2 shows that predicting the object label first prior to generating the part segmentation masks leads to better mask prediction performance”, but this trend does not consistently hold across methods like PLUM (ft), GLaMM (ft), and PixelLM-13B (ft), which contradict the stated conclusion.
- **W9. No Discussion of Results in Figure 4.** The experimental setup and the results shown in Figure 4 are not discussed, leaving the reader unclear about its purpose.
- **W10. Lack of Insightful Analysis in Table 3.** The conclusion that "the proposed experimental setting dampens model-to-model variance" (L265–267) is vague and uninformative. Further analysis is required to understand whether this variance is caused by experimental setup, model architecture, or other confounding factors.

**Clarity:**

- **W11. Unsubstantiated Claim About Training Data Size (L22).** The paper claims that other segmentation-capable LMMs are trained on significantly more data, but no concrete evidence or statistics are provided to support this statement. Details should be included.
- **W12. Missing Metric Descriptions in Table 3.** Table 3 presents several quantitative results without describing the specific calculation evaluation metrics.

**Minor:**

- **L227–228:** The functions $g(\cdot)$ and $N^+$ are introduced without any definitions or explanations.
- All figures should be rendered as vector graphics to enhance clarity and readability, particularly for publication purposes.

---

> ### Author Rebuttal · Authors · 2025-07-31
>
> We sincerely appreciate the time and effort you have dedicated to reviewing our work. In the following, we have provided supplemental details and additional experiment results you requested to address your concerns.
>
> > **W1: Lack of Demo Samples from Partonomy / Partonomy-Core**
>
> We provide Partonomy-Core samples in Figures 1 and 6 and will add more with model predictions.
>
> ---
>
> > **W2: Unverified Claim Regarding Bidirectional Attention**
>
> Prior work shows that bidirectional models (e.g., BERT, BiLSTM-CRF [1-4]) outperform causal ones for token-level tasks, as causal masking is suboptimal [5,6]. Thus, we apply a bidirectional layer for BIO tagging. For example:
>
> 1. Watering hoses are on the ground.
> 2. Watering the grass is important.
>
> Without the second word, it's unclear if "watering" is a compound noun.
>
> [1] BERT: Pre-training of Deep Bidirectional Transformers for Language Understanding
> [2] Bidirectional LSTM-CRF Models for Sequence Tagging
> [3] Deep contextualized word representations
> [4] LUKE: Deep Contextualized Entity Representations
> [5] Acquiring Bidirectionality via Large and Small Language Models
> [6] Looking Right is Sometimes Right: Investigating Decoder-only LLMs for Sequence Labeling
>
> (ii) **Empirical Evidence**
> We train PLUM with/without the bidirectional layer on Partonomy-PartImageNet:
>
> ||b-acc|i-acc|o-acc|
> |-|-|-|-|
> |PLUM (bi)| 98.59 | 87.32 | 99.98 |
> |PLUM (no bi)| 100.00 | 15.86 | 99.78 |
>
> The bidirectional layer is vital for accurate `I` predictions. The model likely uses answer templates to get good `B` accuracy.
>
> On Referring Expression Segmentation (RefCOCO | Unc | Val):
>
> ||b-acc|i-acc|o-acc|
> |-|-|-|-|
> | PLUM (bi) | 99.98 | 99.87 | 100.00 |
> | PLUM (no bi) | 6.68  | 4.92  | 99.98  |
>
> ---
>
> > **W3: Unclear Roles of Mask Feedback Loop Components**
>
> The feedback loop lets prior mask predictions inform current predictions. SAM's mask decoder can take a prior mask (feature map) as a prompt; a natural approach is then to modify SAM's mask encoder to integrate multiple masks.
>
> An initial design might use SAM's mask encoder to embed previously predicted masks (n_masks, h, w, d), fusing them into a single feature map representing previously predicted masks (patch-wise attention to pool into learnable CLS tokens).
>
> However, naively fusing prior masks neglects their semantics. As intuition, one can imagine that if parts are typically disjoint, then prior mask areas should not be predicted in the future. On the other hand, a prior segmentation of an airplane's body can be used to constrain a prediction for windows.
>
> FiLM layers use the textual span embeddings corresponding to masks to modulate their feature maps. [7]'s original usage supports our design. Cross attention is limited by there being one text embedding per mask (it would receive all of the attention weight), but an alternative could be to perform an attention pool by stacking the text embedding with all mask features. We leave this design to future work.
>
>
> [7]: FiLM: Visual Reasoning with a General Conditioning Layer
>
> ---
>
> > **W4: Focal-Tversky Loss Lacks Justification (Impact of DICE vs. Focal-Tversky)**
>
> Focal-Tversky loss (FTL) generalizes DICE: they are equivalent when FTL's coefficients are 0.5. The marginal improvements of FTL over DICE shown below (\alpha=0.7 and \beta=0.3 in Section A.1) are why we select it. PLUM's large performance gains, especially zero-shot, do not stem from FTL. (Evaluation on Identification task of Partonomy-PartImageNet):
>
> | | micro-giou | macro-giou | b-acc | i-acc | o-acc |
> |-|-|-|-|-|-|
> | PLUM (DICE) (ft)| 66.47 | 79.86 | 92.99 | 86.48 | 99.99 |
> | PLUM (FTL) (ft) | 67.90 | 80.30 | 98.59 | 87.32 | 99.98 |
>
> ---
>
> > **W5: Incomplete Ablation of Core Components**
>
> Figure 5 ablates starting from LISA, then adds our span extractor and mask feedback loop incrementally, showing performance on Partonomy-PartImageNet and TextVQA across seven KL-weights.
>
> Plotting all combinations of ±span extractor/±feedback loop is infeasible, but Figure 5 covers 3/4: LISA (–SE, –F), PLUM no feedback loop, and PLUM (both components). At reviewer request, we added LISA + feedback loop. Both components are beneficial.
>
> |Model | micro-gIoU | macro-gIoU|
> |-- | -- | -- |
> |LISA *(-SE, -F)* | 65.6 | 67.7 |
> |LISA + Feedback Loop *(-SE, +F)* | 66.7 | 68.6 |
> |PLUM - Feedback Loop *(+SE, -F)* | 61.4 | 73.9 |
> |PLUM *(+SE, +F)* | **67.9** | **80.3** |
>
> We also ablate the span extractor by removing the bidirectional layer. (ReasonSeg, conducted on epoch=5/25 due to shortage of time):
>
> | | gIoU| cIoU|
> |-|-|-|
> | PLUM (w/ bidirectional)| 41.86 | 34.97 |
> | PLUM (w/o bidirectional)| 37.28 | 31.54 |
>
> ---
>
> > **W6: Lack of Analysis on Loss Weights**
>
> Appendix A.1 shows PLUM uses hyperparameters from LISA, GLaMM, and PixelLM ($\lambda_{CE}=1.0, \,\lambda_{BCE}=2.0$); only $\lambda_{seg}$ and $\lambda_{cls}$ differ. The table below shows the search for our choices:
>
> | $\lambda_{CE}$ | $\lambda_{seg}$ | $\lambda_{BCE}$ | $\lambda_{cls}$ | $\lambda_{KL}$ | lr | gIoU | cIoU |
> |-|-|-|-|-|-|-|-|
> |1.0|2.0|2.0|2.0|0.1|3e-4|0.385|0.331|
> |1.0|4.0|2.0|2.0|0.1|3e-4|0.489|0.506|
> |1.0|5.0|2.0|2.0|0.1|3e-4|0.517|0.494|
> |1.0|6.0|2.0|2.0|0.1|3e-4|0.527|0.501|
> |1.0|8.0|2.0|2.0|0.1|3e-4|0.573|0.546|
>
> We also show early results on ReasonSeg with $\lambda_{cls}$. Increasing $\lambda_{cls}$ leads to slightly better performance.
>
> |$\lambda_{CE}$|$\lambda_{seg}$|$\lambda_{BCE}$|$\lambda_{cls}$|$\lambda_{KL}$|lr|gIoU|cIoU|
> |-|-|-|-|-|-|-|-|
> |1.0|2.0|2.0|0.5|0.5|3e-4|0.110|0.133|
> |1.0|2.0|2.0|1.0|0.5|3e-4|0.103|0.108|
> |1.0|2.0|2.0|2.0|0.5|3e-4|0.127|0.138|
> |1.0|4.0|2.0|0.5|1.0|3e-4|0.239|0.184|
> |1.0|4.0|2.0|2.0|1.0|3e-4|0.263|0.181|
>
> ---
>
> > **W7: No Evaluation of Span Extractor’s BIO Classification**
>
> Please refer to W2 for these results.
>
> ---
>
> > **W8: Misleading Conclusion in L268–273**
>
> We acknowledge this is an unintended misrepresentation of the results. Table 2 shows that predicting objects prior to part masks (Whole2Part) has clear benefits for the pretrained models—all four models have higher micro and macro gIoUs than Part2Whole. This advantage evaporates once the models have seen enough part data, as noted about the finetuned models.
>
> Conditioning also shows clear benefits in Table 3, where knowing the object assists with predicting its parts (Whole2Part's PA >= Part2Whole's PA) and knowing parts helps predict the object (Part2Whole's OA >= Whole2Part's OA).
>
> We will clarify both of these points in the text.
>
> ---
>
> > **W9: No Discussion of Results in Figure 4**
>
> Figure 4 motivates our work (noted on line 201) by showing existing segmenting LMMs perform poorly on part segmentation benchmarks despite training. Finetuning on Partonomy boosts performance, and PLUM outperforms other baselines in zero-shot part segmentation, demonstrating strong generalizability. We will expand the discussion.
>
> ---
>
> > **W10: Lack of Insightful Analysis in Table 3**
>
> One key finding of Table 3 is that prior knowledge of objects increases part selection accuracy (and vice versa). This is nontrivial as we intentionally create difficult wrong answer choices close to the correct answer:
>
> ```
> > "The dirt bicycle in the image has the following visible parts: frame, handlebars, knobby tires/wheels, pedals, and seats.",
> > "The dirt bicycle in the image has the following visible parts: frame, handlebars, lock, and seats.",
> > "The dirt bicycle in the image has the following visible parts: frame, grip, handlebars, knobby tires/wheels, pedals, seats, and tires/wheels.",
> ...
> ```
>
> We highlight this difficulty on line 266 . Answer similarity is reflected by the high precision and recall of the "Random" baseline in Table 3.
>
> Model variance is explained in part by the models' pretraining datasets. PixelLM and GLaMM incorporate their own data into their pretraining mixes (noted in W11 below). These pretraining differences can make a large difference on the zero-shot Partonomy-Core, many of whose parts are not contained in the Partonomy training splits.
>
> Multiple-choice accuracy drops after finetuning, likely due to catastrophic forgetting and low-variance data; next-token prediction alone seems insufficient for part detection, suggesting a future research direction.
>
> ---
>
> > **W11: Unsubstantiated Claim About Training Data Size**
>
> GLaMM and PixelLM are trained on large segmentation datasets: GLaMM on GranD (7.5M concepts, 810M regions) and PixelLM on MUSE (239k instances, 910k regions). We will add these details.
>
> ---
>
> > **W12: Missing Metric Descriptions in Table 3**
>
> We describe the Precision ($P$), Recall ($R$) and Part and Object Accuracy ($PA$ and $OA$) metrics in the caption of Table 3 and on lines 186-191. We use $P$ and $R$ to capture the similarity between the predicted part texts and the ground-truth part texts. $PA$ and $OA$ denote all-or-nothing prediction accuracy.
>
> ---
>
> > **Minor 1: function g and value N+ are undefined**
>
> We appreciate the reviewer’s note for clearer definitions. The function $g$ is a single linear projection layer whose output $q_i$ acts as a query in the SAM decoder. $N+$ refers to the number of positive (B/I) tokens. We will clarify these in the manuscript.
>
> ---
>
> > **Minor 2: Figures should be rendered as vector graphics**
>
> We will rework Figure 4 to use vector graphics or be of higher resolution for clarity.
>
> > **Limitations**
>
> A shortened version of our limitations:
>
> Partonomy-Core has the most part labels among part-grounding benchmarks but omits rare/domain-specific concepts and lacks object variety versus PartImageNet++. Extending to more concepts (e.g., via PartImageNet++) would improve part-level LMM understanding.
> Though PLUM addresses key architectural challenges, part segmentation and identification remain difficult. Autoregressive feedback may accumulate errors and requires sequential decoding.
> As an LLM, PLUM can be misused (e.g., for generating fake or abusive content).

---

> > ### Comment · Reviewer_gK9E · 2025-08-08
> > **Official Response by Reviewer gK9E**
> >
> > Thanks for your detailed responses. Although most of my questions have been addressed through the additional explanation and new experiments, several critical concerns remain unresolved:
> >
> > **W3**: The motivation for leveraging FiLM layers still remains unclear for me. As we know, cross-attention layers can automatically adjust their attention weights by learning from data. Ideally, it can learn to down-weight irrelevant regions and up-weight relevant ones, which appears to serve overlapping purposes of FiLM layers. Please elaborate on the fundamental differences between them and explain why FiLM provides an advantage in this setting.
> >
> > **W4**: Although FTL offers a marginal performance gain, it leads to an unfair comparison to previous works. The motivation for applying extra FTL is weak or unclear.
> >
> > **W10**: The manuscript does not explicitly connect Table 3 to the key finding ("prior knowledge of objects increases part selection accuracy") mentioned above. Please state this point directly when introducing Table 3. Otherwise, readers might conclude that its purpose is only to show “the proposed challenging experimental setting dampens model-to-model variance” (L265–267).

---

> > > ### Comment · Area_Chair_gfz1 · 2025-08-09
> > >
> > > Dear reviewer, as the discussion period is approaching ending, we would be appreciated if you can post feedback at this point especially where Authors ask for such discussions. For your comments that arrive late - Authors may have insufficient time now to address them in depth.

---

> ### Author Response · Authors · 2025-08-09
> **Follow-up Response to Reviewer gK9E**
>
> Thank you for your follow-up questions. We would like to respectfully address each point below.
>
> > **W3: Motivation for FiLM vs. Cross Attention**
>
> Our aim is to reuse SAM’s pretrained mask decoder, which accepts a text embedding and a feature map representing the prior mask. In the feedback loop, this prompt should represent all prior masks' features $m_i \in \mathbb{R}^{h \times w \times d}$ for $i \in [T - 1]$, where each $m_i$ is conditioned on its corresponding text embedding $t_i \in \mathbb{R}^d$.
>
> We would like to respectfully point out that cross attention is ill-suited here: for a mask feature map $m_i$ and its corresponding text embedding $t_i$, we must set $Q = m_i$ to obtain a feature map for the mask prompt (setting $Q = \{t_i\}$ would yield only a single embedding). Therefore, to modulate $m_i$ with $t_i$ we would need to put $K=\{t_i\}$. Computing the attention weights with $QK^T$ then places all the attention weight onto the single text embedding, yielding degenerate outputs where each spatial embedding is the same projection of $t_i$.
>
> FiLM instead directly modulates the feature map $m_i$ via learned affine projections of $t_i$:
>
> $\widetilde{m_i} = \gamma(t_i) \odot m_i + \beta(t_i)$.
>
> This operation preserves spatial structure while injecting text semantics. We then aggregate the set of $\{\widetilde{m_i}\}$ across timesteps via patch-wise attention pooling to form the final prompt. Our use of FiLM achieves our goal to modulate the feature maps with a single text embedding.
>
> > **W4: Fairness and Motivation for FTL**
>
> As we mentioned above, Focal-Tversky loss (FTL) is a generalization of DICE (the two losses are equivalent at $\alpha=\beta=0.5$) and lets us slightly bias towards recall ($\alpha=0.7,\beta=0.3$) for small parts.
>
> We respectfully disagree that using the Focal-Tversky loss to train PLUM is unfair to the other baselines for the following reasons:
>
> 1. As shown in our rebuttal, training with FTL instead of DICE loss improves macro-gIoU by only 0.55%. By contrast, PLUM has an 18.6% higher macro-gIoU than the DICE-trained LISA, showing that FTL is not responsible for the gains (a statement underscored by PLUM's ablations).
>
> 2. The Focal-Tversky loss reweights the impact of precision vs. recall. If FTL were the main driver of PLUM's success on part segmentation, PLUM should have performed worse on non-part-based segmentation tasks, as we traded precision for recall in our FTL hyperparameters. However, this is not the case—PLUM still outperforms LISA on Reasoning Segmentation at both the 7B and 13B model sizes.
>
> Focal-Tversky loss is not our work's contribution, and we show it provides only marginal gains over DICE compared to our core contributions. Regardless, we chose to use FTL to maximize PLUM's performance on part segmentation.
>
>
> > **W10: Explicit Key Finding for Table 3**
>
> We will state explicitly that Table 3 shows prior knowledge of objects increases part selection accuracy (and vice versa), even with closely confusable answer choices.

---

### Note · Authors · 2025-08-15

We would like to thank all of the reviewers for their insightful feedback and the AC for facilitating the review process. We sincerely appreciate the time devoted to writing reviews and engaging in discussions, which have greatly contributed to refining our work.

---

### Decision · Program_Chairs · 2025-09-17

**Decision:**

Accept (spotlight)

**Comment:**

Reviewers think that this paper addresses an important problem, as well as introducing a benchmark which would provides a valuable research topic and resource for the community. After rebuttal and discussion, most of the concerns raised by reviewers are resolved the authors and all the ratings are positive.  I thereby recommend to accept this submission.